# Proteomics reveals synergy between biomass degrading enzymes and inorganic Fenton chemistry in leaf-cutting ant colonies

**Morten Schiøtt\*, Jacobus J Boomsma**

Centre for Social Evolution, Department of Biology, University of Copenhagen, Universitetsparken, Copenhagen, Denmark

**Abstract** The symbiotic partnership between leaf-cutting ants and fungal cultivars processes plant biomass via ant fecal fluid mixed with chewed plant substrate before fungal degradation. Here we present a full proteome of the fecal fluid of *Acromyrmex* leaf-cutting ants, showing that most proteins function as biomass degrading enzymes and that ca. 85% are produced by the fungus and ingested, but not digested, by the ants. Hydrogen peroxide producing oxidoreductases were remarkably common in the proteome, inspiring us to test a scenario in which hydrogen peroxide reacts with iron to form reactive oxygen radicals after which oxidized iron is reduced by other fecal-fluid enzymes. Our biochemical assays confirmed that these so-called Fenton reactions do indeed take place in special substrate pellets, presumably to degrade plant cell wall polymers. This implies that the symbiotic partnership manages a combination of oxidative and enzymatic biomass degradation, an achievement that surpasses current human bioconversion technology.

## Introduction

Mutualistic mergers of simpler biological entities into organizationally complex symbioses have been key steps in the evolution of advanced synergistic forms of life, but understanding the origins and secondary elaborations of such natural cooperative systems remains a major challenge for evolutionary biology (*Smith and Szathmáry, 1997*; *Bourke, 2011*). Physiological complementarity may be a main driver to maintain symbiotic associations, but new adaptations are also expected to jointly exploit the full potential of symbiosis (*Herre et al., 1999*; *Foster and Wenseleers, 2006*). The leaf-cutting ants belong to the genera *Acromyrmex* and *Atta* and form the phylogenetic crown group of the attine fungus-growing ants, which all live in mutualistic symbiosis with fungus-garden symbionts (*Mueller et al., 2005*). While the basal branches of the attine clade farm a considerable diversity of mostly leucocoprinaceous fungi, normally in underground chambers (*Branstetter et al., 2017*), the leaf-cutting ants almost exclusively farm what has been described as the highly variable cultivar species *Leucocoprinus gongylophorus* (*Schultz and Brady, 2008*). The ants cut leaf fragments from the canopy and understory and bring them back to their nest to be processed as growth substrate for the cultivar. This involves chewing forage material into smaller pieces, which are mixed with fecal fluid and deposited as new substrate on the actively growing parts of the garden (*Weber, 1966*).

The deep ancestors of the attine ants and their fungal cultivars were hunter-gatherers and saprotrophs (*Schultz and Brady, 2008*), respectively, which implies that none of them were adapted for efficiently degrading fresh plant material. Neither was there any need initially, because all phylogenetically basal attine ants provision their gardens with dead plant debris. There is a striking difference between the small colonies of these basal attine ants and the large colonies of leaf-cutting ants which are conspicuous functional herbivores with a large ecological footprint (*Mehdiabadi and*

**\*For correspondence:**
mosch@aqua.dtu.dk

**Competing interests:** The authors declare that no competing interests exist.

**eLife digest** Colonies of tropical leaf-cutting ants live in underground nests where a fungus grows that feeds them. The ants, in turn, provide the fungus with the freshly-cut leaf fragments it needs for nutrition. The relationship between the ants and the fungus, in which they live close together and help one another survive, is known as symbiosis. It is an ancient, extremely well integrated relationship, in which neither species can survive without the other. However, the details of how the ants and the fungus work together to break down the leaf fragments so they can be used for nutrition are not well understood.

When the ants eat the fungus, they do not digest its enzymes (the proteins that accelerate chemical reactions in a cell). Instead, the fungal enzymes travel through the ants' gut and into their fecal liquid, which gets deposited on the fresh-cut leaves when the ants collect them. The ants then make temporary pellets out of the new leaf fragments before providing them to the fungus. To better understand how each species contributes to the breakdown of the leaf fragments, Schiøtt and Boomsma identified all the proteins present in the fecal fluid of the ants. Once they had a complete list of about 100 proteins, they determined which of them were produced by the fungus and which by the ant. Schiøtt and Boomsma observed that certain combinations of fungal and ant enzymes could trigger a Fenton reaction – a chemical reaction that efficiently begins the breakdown of the tough walls around plant cells. This reaction is so aggressive that it is rarely found in nature, but it could help explain the high efficiency of the fungus and the ants symbiotically processing leaf fragments.

But could a Fenton reaction actually proceed in the ants' nest without hurting the ants or affecting the rest of the fungal garden? The evidence obtained suggested that the temporary pellets made by the ants serve to isolate the reaction, so the aggressive chemistry takes place away from the ants and detached from the fungal gardens.

Schiøtt and Boomsma showed that the symbiotic relationship between the ants and the fungus has led to a sustainable and efficient way of breaking down plant materials to use them for nutrition. The Fenton reaction is economically important in many industries, including bioethanol production, the detergent industry, and food production. Emulating the methods used by leaf-cutting ants, which have been fine-tuned by millions of years of natural selection, may allow humans to develop more efficient technologies for breaking down organic compounds.

Schultz, 2010; Mueller, 2002; Shik et al., 2021). However, most of the leaves that the ants harvest are heavily protected against herbivory by recalcitrant cell walls and toxic substances (Chen, 2008), defenses that needed to be overcome for obtaining net nutrition. In recent decades, a substantial amount of research has elucidated the selection forces that shaped the functional herbivory niche of the leaf-cutting ants. It was shown that: 1. the 'higher' attine ants to which the leaf-cutting ants belong rear specialized, gongylidia-producing cultivars that lost their free-living relatives and could thus begin to co-evolve with the ants (Schultz and Brady, 2008); 2. the leaf-cutting ant cultivar *Leucocoprinus gongylophorus* evolved to be obligatorily multinucleate and polyploid, which may well have allowed higher crop productivity (Kooij et al., 2015a); 3. leaf-cutting ant queens evolved to be multiply inseminated, in contrast to all phylogenetically more basal attines studied; this meant that not only the garden symbiont but also the farming ant families became genetically 'chimeric', albeit not as cell-lines but as lines of fungal nuclei and individual patrilines of ants (Villesen et al., 2002); 4. the leaf-cutting ants evolved distinct division of labor within the worker caste, with small, medium, and large workers and specialized soldiers (the latter only in *Atta*) (Weber, 1966); and 5. the crown group of the attine ants underwent significant changes in their gut microbiota that likely facilitated the conversion efficiency of fungal diet substrate into ant biomass (Sapountzis et al., 2015; Sapountzis et al., 2019).

While these advances have significantly enriched our understanding of the spectacular biology of the attine ants in general and the leaf-cutting ants in particular, very few studies have addressed the molecular synergy mechanisms that were decisive for integrating the complementary innovations of the ants and their cultivars. Recent genome sequencing showed that cultivars likely changed their chitinase processing abilities in co-evolution with the ants' labial glands (Nygaard et al., 2016), but

the clearest and most detailed evidence of functional physiological complementarity has been found for the digestive enzymes of the mutualistic farming symbiosis. These insights were initiated by pioneering work by Michael M. Martin (*Boyd and Martin, 1975b*; *Boyd and Martin, 1975a*; *Martin and Martin, 1971*; *Martin and Martin, 1971*), which suggested that proteases in the fecal fluid of *Atta* leaf-cutting ants had actually been produced by the fungal symbiont and passed unharmed through the ant guts to mediate protein degradation of the plant forage material when it was mixed with fecal fluid. Our previous work not only confirmed this hypothesis for Acromyrmex leaf-cutting ants (*Kooij et al., 2014b*), but also expanded the list of cultivar-derived fecal fluid proteins to include pectinases, hemicellulases, and laccases (*Rønhede et al., 2004*; *Schiøtt et al., 2010*; *De Fine Licht et al., 2013*; *Kooij et al., 2016*). These studies also showed that all these enzymes are produced in the specialized gongylidia of the fungal cultivar that the ants ingest as their almost exclusive food source (*Schiøtt et al., 2010*; *De Fine Licht et al., 2013*; *Kooij et al., 2014b*; *Kooij et al., 2016*).

While these recent studies generalized Martin's findings across the two leaf-cutting ant genera and for multiple groups of enzymes, they left a number of deeper questions regarding the symbiotic decomposition process unanswered. First, the degradation of leaf fragments in *Acromyrmex* fungus gardens proceeds very efficiently, as ant-produced pellets of chewed leaf material deposited on top of the garden turn black within a few hours to then be spread out over the garden by the ants. Second, the fungal cultivar has lost a major ligninase domain (*Nygaard et al., 2016*), raising the question how these recalcitrant (difficult-to-degrade) cell wall components are breached to allow fungal hyphae access to the nutritious interior of live plant cells. To address what processes we might have missed, the present study focused on obtaining a full proteome of the fecal fluid of the leaf-cutting ant *Acromyrmex echinatior*, extending the earlier partial proteome from 33 to 101 identified proteins (*Schiøtt et al., 2010*). This revealed a surprisingly high proportion of oxidoreductases, which made us conjecture that the mutualistic ant and fungal partners might jointly realize aggressive biomass conversion via the inorganic Fenton reaction, which produces hydroxyl radicals that can break down lignocellulose. We thus set up a series of experiments to investigate whether Fenton chemistry indeed takes place when fecal fluid interacts with chewed leaf fragments concentrated in the ca. 3 mm diameter pellets that the ants construct. We confirmed our basic hypothesis and also found other enzymatic and iron components of the fecal fluid participating in the recycling of Fenton chemistry reactants. The synergistic process of enzymatic and inorganic chemistry appears to sustain a continuous production of hydroxyl radicals until the ants terminate the process. They do that by dismantling their Fenton bioconversion pellets to offer the pre-digested plant substrate to the fungal hyphae as small fragments and seemingly without any collateral damage to ant or fungal tissues due to uncontrolled hydroxyl radicals.

## Results

### Characterization of the fecal fluid proteome

Our proteomic analysis of worker fecal fluid from four different *A. echinatior* colonies identified 174 proteins (*Supplementary file 1*), which showed substantial overlap among the pooled colony-specific samples (*Figure 1A*). To avoid including spurious proteins, we restricted our analyses to a shortlist of 101 proteins present in at least three of the four colony samples, and we were able to annotate 91 (90%) of these proteins (*Table 1*). This core proteome consisted mainly of degradation enzymes belonging to five major functional categories: oxidoreductases, proteolytic enzymes, carbohydrate active enzymes (CAZymes), phosphate-liberating enzymes, and lipid degrading enzymes (lipidolytic enzymes) (*Figure 1B*). Of the 10 remaining proteins, only three could not be functionally annotated. The vast majority (86) of the core fecal proteome originated from the fungal symbiont, with only 15 being encoded by the ant genome. Based on the number of spectra obtained for each identified protein, MaxQuant produced an approximate measure of relative abundance of each protein called label-free quantification intensity (LFQ intensity) (*Table 1*). This revealed that lipidolytic enzymes constitute only a very small fraction and that the proteins in the 'miscellaneous' category collectively had a moderate representation although some of these proteins had high abundance. The four remaining categories had summed LFQ intensities of the same order of magnitude (*Figure 1B*) and two specific proteins stood out as being exceptionally abundant

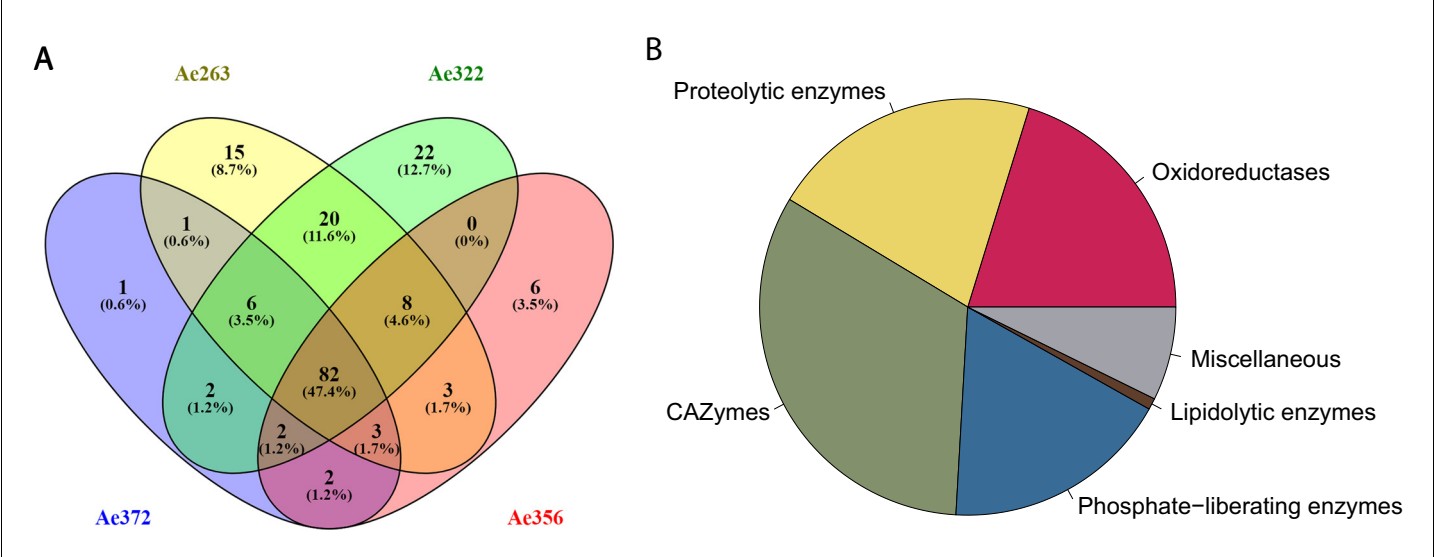

**Figure 1.** Statistics of the fecal fluid proteome of *Acromyrmex echinatior*. (A) Venn diagrams, constructed using the web application Venny 2.1 (http://bioinfogp.cnb.csic.es/tools/venny/index.html) showing the overlap of protein profiles identified in the four fecal fluid samples obtained from colonies Ae263, Ae322, Ae356, and Ae372. (B) Pie chart showing the abundances of proteins across colonies assigned to six functional categories based on the label-free quantification (LFQ) values provided by MaxQuant.

(*Supplementary file 1*), the Alkaline phosphatase (Protein ID 75 in *Table 1*) and the previously described Laccase LgLcc1 (Protein ID 97 in *Table 1*; *De Fine Licht et al., 2013*), which jointly made up almost a quarter of the total fecal protein mass. Three other proteins also showed remarkably high LFQ-abundance levels: a 1,3-beta-glucanase (Protein ID 88 in *Table 1*), a Metallopeptidase M35 (Protein ID 206 in *Table 1*), and a Serine protease S53 (Protein ID 64 in *Table 1*), which implies that the five most abundant proteins contributed more than 40% of the total fecal protein mass. These results confirmed our previous findings that a large proportion of the fecal fluid proteins consists of proteases (*Kooij et al., 2014b*), pectinases (*Schiøtt et al., 2010*), hemicellulases (*Kooij et al., 2016*), and laccases (*De Fine Licht et al., 2013*), but we now also identified several phosphate-liberating and lipidolytic enzymes.

## The complementary components needed for Fenton chemistry

Five of the identified oxidoreductases were annotated as fungal glucose-methanol-choline (GMC) oxidoreductases. A subgroup of these enzymes are the aryl alcohol oxidases that are believed to play an important role in lignin degradation by brown-rot fungi (*Guillén et al., 2000*). This category of wood degrading fungi is distinct from the white-rot fungi, which degrade lignin using ligninolytic peroxidases (*Arantes et al., 2012*; *Arantes and Goodell, 2014*). The latter mechanism has most likely evolved only once in the Agaricomycotina lineage (to which *Leucocoprinus* belongs), but has subsequently been lost many times giving a complex phylogenetic pattern (*Floudas et al., 2012*). We showed previously (*Nygaard et al., 2016*) that the leaf-cutting ant cultivar has lost its ligninolytic peroxidases, so their decomposition spectrum would functionally be more reminiscent of brown-rot fungi than white-rot fungi although they are not wood degrading fungi and are thus not formally covered by this terminology. Assuming that convergent adaptation could have taken place in the *Leucocoprinus* symbiont, we thus pursued the pathways that could enable the Fenton reaction in more detail.

In order to break down lignocellulose, brown-rot fungi produce aromatic alcohols, which are subsequently oxidized by aryl alcohol oxidases to produce hydrogen peroxide ($H_2O_2$). The hydrogen peroxide is subsequently used in a Fenton reaction in which reduced iron ($Fe^{2+}$) reacts with hydrogen peroxide to produce hydroxyl radicals that can diffuse into the lignocellulose substrate and break the chemical bonds responsible for the rigid structure of lignocellulose. This in turn will loosen the cell walls and allow access for more bulky degradation enzymes to target specific components of the plant cell walls. Unless the oxidized iron ($Fe^{3+}$) is recycled by reduction to $Fe^{2+}$, the production

**Table 1.** Fecal fluid proteins that were found in three or four of the examined colony samples (*Figure 1*), and thus likely to belong to the core fecal fluid proteome.

For each protein the predicted function, its fungal or ant origin, the number of samples containing the protein, the relative abundance of the protein, and an identifier number are listed. Proteins were assigned to one of six functional categories (*Figure 1*). Although a number of oxidoreductases are listed in the CAZy database in the subcategory of auxiliary activities we have for the present study kept them separate. This is because it remains ambiguous whether they are true CAZymes if they are defined as redox enzymes that 'act in conjunction with CAZymes'. This separation also resolved the problem that some of the oxidoreductases listed in Table 1 are not covered by the CAZy database. Question marks indicate that annotations were inconclusive.

| Category | Protein ID | Predicted function | Source | Samples with protein | Relative abundance |
|---|---|---|---|---|---|
| Oxidoreductases (including CAZymes with auxilliary activity) | 15 | Laccase | Fungus | 4 | 106.0 |
| | 97 | Laccase | Fungus | 4 | 431.8 |
| | 180 | Laccase | Fungus | 4 | 14.7 |
| | 184 | Laccase | Fungus | 4 | 21.4 |
| | 35 | 4-carboxymuconolactone decarboxylase, alpha-beta hydrolase | Fungus | 4 | 4.3 |
| | 81 | Copper radical oxidase, glyoxal oxidase, galactose oxidase | Fungus | 4 | 8.9 |
| | 186 | Copper radical oxidase, glyoxal oxidase, galactose oxidase | Fungus | 4 | 8.3 |
| | 98 | FAD/FMN-containing isoamyl alcohol oxidase | Fungus | 4 | 41.1 |
| | 9 | GMC oxidoreductase, aryl-alcohol oxidase | Fungus | 4 | 5.9 |
| | 86 | GMC oxidoreductase, aryl-alcohol oxidase | Fungus | 4 | 10.3 |
| | 178 | GMC oxidoreductase, aryl-alcohol oxidase | Fungus | 4 | 6.9 |
| | 191 | GMC oxidoreductase, aryl-alcohol oxidase | Fungus | 4 | 18.3 |
| | 23 | GMC oxidoreductase, aryl-alcohol oxidase | Fungus | 3 | 1.9 |
| | 145 | GMC oxidoreductase, glucose dehydrogenase | Ant | 4 | 82.2 |
| | 149 | Succinate dehydrogenase (ubiquinone) flavoprotein subunit | Ant | 3 | 4.8 |
| Proteolytic enzymes | 6 | Aspartic peptidase A1A, polyporopepsin | Fungus | 4 | 12.0 |
| | 111 | Aspartic peptidase A1A, saccharopepsin | Fungus | 4 | 74.3 |
| | 125 | Metallopeptidase M1, ERAP2 aminopeptidase | Ant | 4 | 11.3 |
| | 112 | Metallopeptidase M14A, zinc carboxypeptidase | Ant | 4 | 27.4 |
| | 144 | Metallopeptidase M14A, zinc carboxypeptidase | Ant | 4 | 7.3 |
| | 141 | Metallopeptidase M28D, carboxypeptidase Q | Ant | 4 | 15.9 |
| | 188 | Metallopeptidase M35, deuterolysin | Fungus | 4 | 23.9 |
| | 206 | Metallopeptidase M35, peptidyl-lys metallopeptidase | Fungus | 4 | 218.6 |
| | 207 | Metallopeptidase M35, peptidyl-lys metallopeptidase | Fungus | 4 | 11.4 |
| | 56 | Metallopeptidase M36 | Fungus | 4 | 4.3 |
| | 133 | Serine protease S1A, chymotrypsin | Ant | 4 | 56.0 |
| | 134 | Serine protease S1A, chymotrypsin | Ant | 4 | 3.8 |
| | 135 | Serine peptidase S1A, chymotrypsin-2 | Ant | 4 | 8.8 |
| | 36 | Serine peptidase S8A, cuticle degrading peptidase | Fungus | 4 | 7.2 |
| | 103 | Serine peptidase S8A, cerevisin | Fungus | 4 | 51.6 |
| | 16 | Serine peptidase S10, carboxypeptidase | Fungus | 3 | 2.0 |
| | 14 | Serine peptidase S10, carboxypeptidase | Fungus | 4 | 1.3 |
| | 24 | Serine peptidase S10, carboxypeptidase OcpA | Fungus | 4 | 31.5 |
| | 95 | Serine peptidase S10, serine-type carboxypeptidase F | Fungus | 4 | 11.7 |
| | 199 | Serine peptidase S28, carboxypeptidase | Fungus | 4 | 6.3 |
| | 33 | Serine peptidase S53, grifolisin | Fungus | 4 | 20.5 |
| | 64 | Serine protease S53, grifolisin | Fungus | 4 | 192.4 |

*Table 1 continued on next page*

*Table 1 continued*

| Category | Protein ID | Predicted function | Source | Samples with protein | Relative abundance |
|---|---|---|---|---|---|
| Phosphate-liberating enzymes | 74 | Acid phosphatase, 3-phytase A | Fungus | 3 | 7.2 |
| | 0 | Acid phosphatase, nucleotidase | Fungus | 4 | 16.9 |
| | 61 | Acid phosphatase, nucleotidase | Fungus | 4 | 24.8 |
| | 75 | Alkaline phosphatase | Fungus | 4 | 432.8 |
| | 100 | Alkaline phosphatase | Fungus | 4 | 14.8 |
| | 109 | Phytase esterase-like | Fungus | 4 | 105.1 |
| | 59 | Phytase esterase-like | Fungus | 3 | 19.2 |
| | 193 | Phytase, histidine acid phosphatase domain | Fungus | 4 | 6.5 |
| | 187 | Phytase, phosphoglycerate mutase, histidine acid phosphatase | Fungus | 4 | 25.2 |
| | 72 | PLC-like phosphodiesterase | Fungus | 3 | 3.5 |
| | 37 | Ribonuclease T1 | Fungus | 4 | 5.6 |
| | 34 | Ribonuclease T2 | Fungus | 4 | 11.5 |
| | 73 | GH3, beta-glucosidase | Fungus | 4 | 52.0 |
| | 99 | GH3, beta-xylosidase | Fungus | 4 | 76.0 |
| | 55 | GH5, endocellulase | Fungus | 3 | 3.7 |
| | 179 | GH5, exo-1,3-beta-glucanase | Fungus | 4 | 9.8 |
| | 183 | GH5, mannan endo-1,4-beta-mannosidase F | Fungus | 4 | 7.6 |
| | 13 | GH5, mannan endo-1,4-beta-mannosidase | Fungus | 3 | 11.7 |
| | 38 | GH10, endo-1,4-beta-xylanase | Fungus | 3 | 3.9 |
| Carbohydrate active enzymes | 121 | GH12, xyloglucanase | Fungus | 3 | 4.8 |
| | 132 | GH13, alpha-glucosidase, maltase | Ant | 3 | 2.9 |
| | 123 | GH15, glucoamylase | Fungus | 4 | 66.5 |
| | 88 | GH17, 1,3-beta-glucanase | Fungus | 4 | 290.8 |
| | 8 | GH18, chitinase | Fungus | 3 | 16.8 |
| | 66 | GH20, betahexosaminidase | Fungus | 3 | 1.3 |
| | 214 | GH25?, lysozyme | Fungus | 4 | 102.4 |
| | 25 | GH27, alpha-galactosidase | Fungus | 4 | 27.9 |
| | 83 | GH28, endo-polygalacturonase | Fungus | 4 | 83.3 |
| | 80 | GH29, alpha-L-fucosidase | Fungus | 3 | 1.8 |
| | 104 | GH31, alpha-glucosidase | Fungus | 4 | 39.1 |
| | 157 | GH31, alpha-glucosidase | Ant | 4 | 7.9 |
| | 108 | GH35, beta-galactosidase | Fungus | 4 | 13.6 |
| | 154 | GH37, trehalase | Ant | 4 | 19.8 |
| | 90 | GH43, arabinan-endo-1,5-alpha-L-arabinosidase | Fungus | 4 | 21.8 |
| | 11 | GH51, arabinofuranosidase | Fungus | 4 | 85.7 |
| | 46 | GH53, arabinogalactanase | Fungus | 4 | 36.2 |
| | 77 | GH55, exo-1,3-beta-glucanase | Fungus | 4 | 34.5 |
| | 7 | GH78, α-L-rhamnosidase | Fungus | 4 | 14.0 |
| | 63 | GH79, betaglucoronidase | Fungus | 4 | 35.0 |
| | 62 | GH88, glucuronyl hydrolase | Fungus | 4 | 3.0 |
| | 76 | GH92, exo-alpha-mannosidase? | Fungus | 4 | 16.6 |
| | 68 | GH92, exo-alpha-mannosidase? | Fungus | 3 | 3.8 |
| | 57 | CE8, pectinesterase | Fungus | 4 | 69.0 |
| | 40 | CE12, rhamnogalacturonan acetylesterase | Fungus | 4 | 5.8 |

*Table 1 continued on next page*

*Table 1 continued*

| Category | Protein ID | Predicted function | Source | Samples with protein | Relative abundance |
|---|---|---|---|---|---|
| | 67 | CE12, Rhamnogalacturonan acetylesterase | Fungus | 4 | 27.6 |
| | 208 | PL1, pectate lyase | Fungus | 4 | 34.6 |
| | 122 | PL4, rhamnogalacturonan lyase | Fungus | 4 | 9.6 |
| Lipidolytic enzymes | 124 | Lipase, triacylglycerol lipase | Fungus | 4 | 2.5 |
| | 65 | Neutral/alkaline nonlysosomal ceramidase | Fungus | 4 | 6.2 |
| | 138 | Pancreatic lipase-related protein | Ant | 4 | 6.5 |
| | 44 | Phosphatidylglycerol/phosphatidylinositol transfer protein | Fungus | 3 | 2.8 |
| | 32 | Alpha/beta hydrolase, cephalosporin esterase | Fungus | 4 | 10.1 |
| | 18 | Alpha/beta hydrolase, triacylglycerol lipase, carotenoid ester lipase | Fungus | 4 | 5.5 |
| | 27 | Carboxylesterase; alpha-beta hydrolase; lipase | Fungus | 4 | 1.5 |
| Miscellaneous proteins | 212 | Fruit-body specific protein D | Fungus | 4 | 44.0 |
| | 31 | Plant expansin, papain inhibitor | Fungus | 3 | 1.7 |
| | 129 | Regucalcin | Ant | 4 | 5.8 |
| | 113 | SnodProt1, cerato platanin | Fungus | 4 | 114.7 |
| | 10 | Ubiquitin | Fungus | 4 | 1.7 |
| | 136 | Epididymal secretory protein E1 | Ant | 4 | 6.4 |
| | 158 | CEN-like protein 2, OV-16 antigen | Fungus | 4 | 4.8 |
| | 106 | Hypothetical protein | Fungus | 4 | 39.2 |
| | 195 | Hypothetical protein | Fungus | 3 | 4.5 |
| | 119 | Hypothetical protein, symbiosis related protein, MAPK? | Fungus | 4 | 52.8 |

of reactive oxygen species will end when all $Fe^{2+}$ has been used. It has therefore been suggested that fungi secrete secondary metabolites such as hydroquinones to reduce $Fe^{3+}$ to $Fe^{2+}$ (*Suzuki et al., 2006*). In spite of the decomposition potential of the Fenton reaction, it remains a major challenge for any living tissues to be exposed to concentrated doses of free radicals. Numerous adaptations have in fact evolved to remove low concentrations of free radicals from living tissues (*Matés et al., 1999*), so we conjectured that attine ants should have found ways to avoid such damage in case they would use Fenton chemistry.

Against this background knowledge from brown-rot fungi, we drew up a hypothetical scenario (*Figure 2*) that might allow the leaf-cutting ant symbiotic partnership to use Fenton chemistry, to sustain and control that reaction according to need, and to have it take place in a spatial setting where collateral damage to living tissues would be avoided. As it appears, the spatial issue appears to have been resolved because *Acromyrmex* colonies concentrate their chewed leaf fragments in pellets of ca. 3 mm diameter at the top of their fungus gardens for some hours before the material is dispersed across the actively growing ridges of the fungal mycelium mass. However, if these pellets would be the optimal location for a sustained Fenton reaction, the symbiotic partnership would have been required to also evolve a fecal fluid enzyme that reduces $Fe^{3+}$ while decomposing degradation products from the plant substrate. We hypothesized that glucose dehydrogenase (Protein ID 145 in *Table 1*), the most abundant ant-encoded fecal fluid protein (*Table 1*, *Supplementary file 1*), could have this function using glucose as electron donor to reduce $Fe^{3+}$, either directly or indirectly via an intermediate redox compound (*Figure 2*). We therefore set up a number of biochemical assays to test the proposed scenario and report the results below.

## Hydrogen peroxide production by fungal GMC oxidoreductases in ant fecal fluid

A phylogenetic tree of all five fungal GMC oxidoreductases isolated from fecal fluid, together with representative GMC oxidoreductases from various basidiomycete fungi, showed that these five enzymes clustered within the aryl alcohol oxidases (*Figure 3*), while being distinct from the four

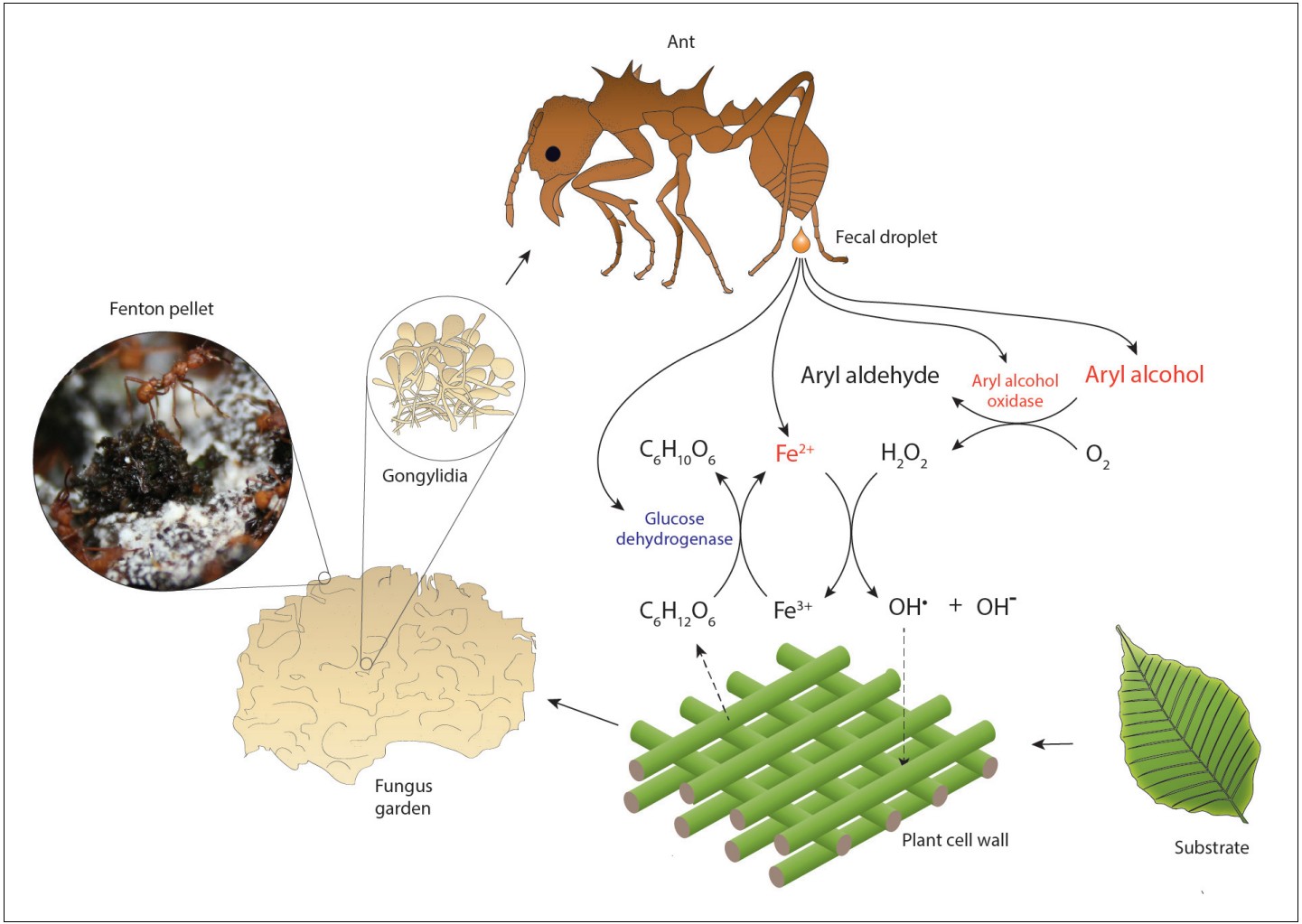

**Figure 2.** Hypothesized reaction scheme for the generation of hydroxyl radicals when ant fecal fluid interacts with chewed leaf fragments in temporary substrate pellets, based on the presence of enzymes in the total fecal fluid proteome (*Supplementary file 1*). Fungal enzymes produced in gongylidia of the symbiotic garden-cultivar that the ants ingest pass unharmed through the gut to end up in the fecal fluid (*Boyd and Martin, 1975b*; *Kooij et al., 2014b*; *Schiøtt et al., 2010*; *De Fine Licht et al., 2013*; *Kooij et al., 2016*). After droplets of fecal fluid are deposited and become exposed to oxygen, the fungal oxidoreductases produce hydrogen peroxide while aryl alcohols are converted to aryl aldehydes. The hydrogen peroxide then reacts with reduced iron ($Fe^{2+}$) to produce hydroxyl radicals ($OH^{\bullet}$) in a Fenton reaction, which aggressively breaks down cell walls of the plant substrate. Oxidized iron ($Fe^{3+}$) can then be reduced again by ant-encoded glucose dehydrogenase, using glucose released via plant cell wall decomposition as electron donor. The leaf substrate is initially concentrated in green pellets of ca. 3 mm diameter distributed across the top of fungus gardens, which turn black in a few hours when subjected to Fenton-mediated degradation (inset image). Compounds ultimately derived from the fungal symbiont are in red text and compounds directly produced by the ants in blue.

other known functional groups (glucose oxidases, methanol oxidases, pyranose 2-oxidases, and cellulose dehydrogenases) (*Ferreira et al., 2015*). We subsequently measured the amount of hydrogen peroxide that these GMC oxidoreductases produce with and without substrates, which showed that fecal fluid contained a substantial amount of hydrogen peroxide. Addition of aromatic veratryl alcohol further increased this amount, whereas glucose and methanol did not have any effect (*Figure 4A*), confirming the presence of aryl alcohol oxidases in the fecal fluid. As an extra control assay to check that the increase in absorbance was directly caused by hydrogen peroxide, we added the enzyme catalase, which is a very effective and specific hydrogen peroxide degrading enzyme. As expected, catalase completely removed the signal, confirming that hydrogen peroxide is indeed naturally produced in fecal fluid (*Figure 4A*). Two other fecal fluid oxidoreductases (Protein IDs 81 and 186 in *Table 1*) were annotated as copper radical oxidases with close similarity to glyoxal oxidases.

These enzymes may also produce hydrogen peroxide, using the small organic compound glyoxal and other aldehydes as substrate (*Daou and Faulds, 2017*). Experimental addition of glyoxal to fecal fluid of *Acromyrmex* worker did indeed increase the amount of hydrogen peroxide, confirming that glyoxal oxidases are also present in the fecal fluid (*Figure 4A*).

## Does ant fecal fluid contain sufficient iron to maintain Fenton chemistry?

For the Fenton reaction to take place, the produced hydrogen peroxide must react with $Fe^{2+}$ atoms. Brown-rot fungi are believed to be able to actively increase the concentration of iron atoms in their wood substrate for this purpose (*Kirker et al., 2017*), but such a mechanism would not be applicable to the fecal fluid Fenton reaction since it is separated from the fungal hyphae. To be viable, this reaction must therefore rely on iron present in the leaf material collected by the ant workers and in the fecal fluid itself. Making fungal iron available would be adaptive for the ant farming symbiosis as a whole because direct use of iron from live plant tissues might well be constrained because plants also produce iron chelator compounds to inhibit the growth of plant pathogens (*Dellagi et al., 2005*). We measured the iron levels in fecal fluid and found an average (± SD) level of $370 ± 110 \mu M$ iron across six ant colonies. This concentration of iron should be sufficient to allow the Fenton reaction to occur (*Hegnar et al., 2019*; *Elias and Waterhouse, 2010*). It corresponds to the levels relevant for natural wood degrading systems (*Hegnar et al., 2019*) although it is one to two orders of magnitude lower than the levels typically used for pretreatment of plant biomass in vitro (*Jung et al., 2015*). However, in such human bioconversion experiments, oxidized iron is not recycled into reduced iron, which means that the pool of useable iron is continuously decreasing and therefore needs a high starting concentration. In living systems there are limits to how much iron can be accumulated without posing a risk for unintended oxidation of organic compounds. Natural selection producing a system for iron recycling could thus be highly adaptive.

## Are fecal fluid hydroxyl radicals produced by Fenton chemistry?

The hydroxyl radicals that Fenton chemistry should produce can be measured via their ability to break down 2-deoxy-D-ribose into thiobarbituric acid-reactive substances (TBARS) (*Halliwell et al., 1988*). To quantify this effect, we adapted the standard 2-deoxy-D-ribose assay to accommodate the small volumes of fecal fluid that could practically be collected directly from the ants and found pronounced amounts of hydroxyl radicals to be present in the ant fecal fluid (*Figure 4B*). Adding a strong metal chelator 1,10-phenanthroline caused a significant reduction in absorbance. This is consistent with the hydroxyl radicals being produced by Fenton chemistry because the Fenton reaction is known to be inhibited by 1,10-phenanthroline's strong iron sequestering activity (*Winterbourn, 1995*). This effect was not observed when only the solvent of the 1,10-phenanthroline chemical (methanol) was added.

## Is iron reduction controlled by the ant encoded glucose hydrogenase?

The Fenton reaction can only proceed as long as there are $Fe^{2+}$ atoms present to be converted to $Fe^{3+}$ atoms, and brown-rot fungi are believed to maintain their Fenton reactions via enzyme systems that convert $Fe^{3+}$ back to $Fe^{2+}$ (*Guillén et al., 2000*; *Arantes et al., 2012*). A similar mechanism would not work for the initial stages of leaf degradation in the fungus garden, as the mycelium is separated from the Fenton pellets. We therefore conjectured that the ant-encoded glucose dehydrogenase identified in the fecal fluid proteome could mediate the reduction of $Fe^{3+}$ to $Fe^{2+}$ using glucose as electron donor. We indeed found a measurable capacity for $Fe^{3+}$ reduction in fecal fluid, which was further increased by the addition of glucose. This was consistent with $Fe^{3+}$ being actively recycled into $Fe^{2+}$ via fecal fluid compounds and with glucose dehydrogenase most likely being responsible for at least part of this activity (*Figure 4C*). An alternative pathway for $Fe^{3+}$ reduction is mediated by laccase (*Guillén et al., 2000*), which was also found in high quantities in the fecal fluid. Preliminary experiments using laccase substrates in these assays did, however, not change the rate of $Fe^{3+}$ reduction. This makes it less likely that laccase plays a role in this process, although more elaborate experimentation will be needed in order to rule out this possibility.

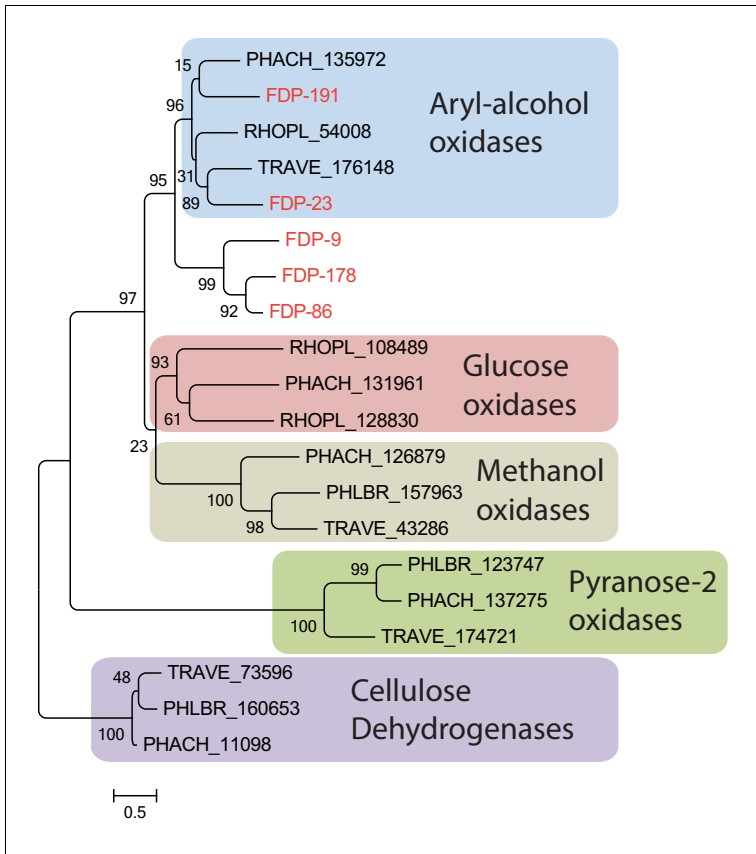

**Figure 3.** Gene tree of the five fungal glucose-methanol-choline (GMC) oxidoreductases identified in the ant fecal fluid (present study) and in representative other basidomycete fungi, all assigned to functional groups based on a previous study (*Ferreira et al., 2015*). All fungal GMC oxidoreductases from the fecal fluid (Protein IDs 9, 23, 86, 178, and 191 in *Table 1*; red text) clustered among the known aryl-alcohol oxidases. Other closely related functional groups are glucose oxidases, methanol oxidases, pyranose-2 oxidases, and cellulose dehydrogenases, which were retrieved in *Phanerochaete chrysosporium* (PHACH), *Rhodonia placenta* (RHOPL), *Trametes versicolor* (TRAVE), and *Phlebia brevispora* (PHLBR), but not in fecal fluid of the leaf-cutting ant *A. echinatior*. Note that the ant-encoded glucose dehydrogenase is also a GMC oxidoreductase, but would sequence-wise not fit into this phylogeny of fungal proteins. Numbers are aLRT SH-like support values for nodes. The scale bar represents 0.5 substitutions per site.

The online version of this article includes the following source data for figure 3:

**Source data 1.** Amino acid sequences used for the gene tree shown in *Figure 3*.

## Discussion

### *Acromyrmex* fecal fluid has a highly specialized proteome

The presence of fungal enzymes in the fecal fluid of leaf-cutting ants was suggested already in the 1970s based on the biochemical similarity of proteases in fecal fluid and the fungal cultivar (*Boyd and Martin, 1975b*; *Boyd and Martin, 1975a*). We achieved increasing confirmation of this idea in a series of previous studies (*Schiøtt et al., 2010*; *Kooij et al., 2014b*; *De Fine Licht et al., 2013*; *Kooij et al., 2016*), but our present study presents the first overall proteome that is complete enough to allow inferences of hitherto unknown chemical processes. The comprehensive list of *A. echinatior* fecal fluid proteins that we obtained (*Supplementary file 1*) was now also sufficiently replicated to confirm that the large majority of the identified proteins are fungal in origin and serve the farming symbiosis after being deposited on new plant substrate via the ant fecal fluid. When run on SDS-Page gels (*Schiøtt et al., 2010*), the fecal fluid proteins always gave the same banding pattern, rejecting the null hypothesis that they are a random collection of proteins that happen to escape digestion. Overall, 101 of the 174 identified proteins were found in at least three fecal fluid samples

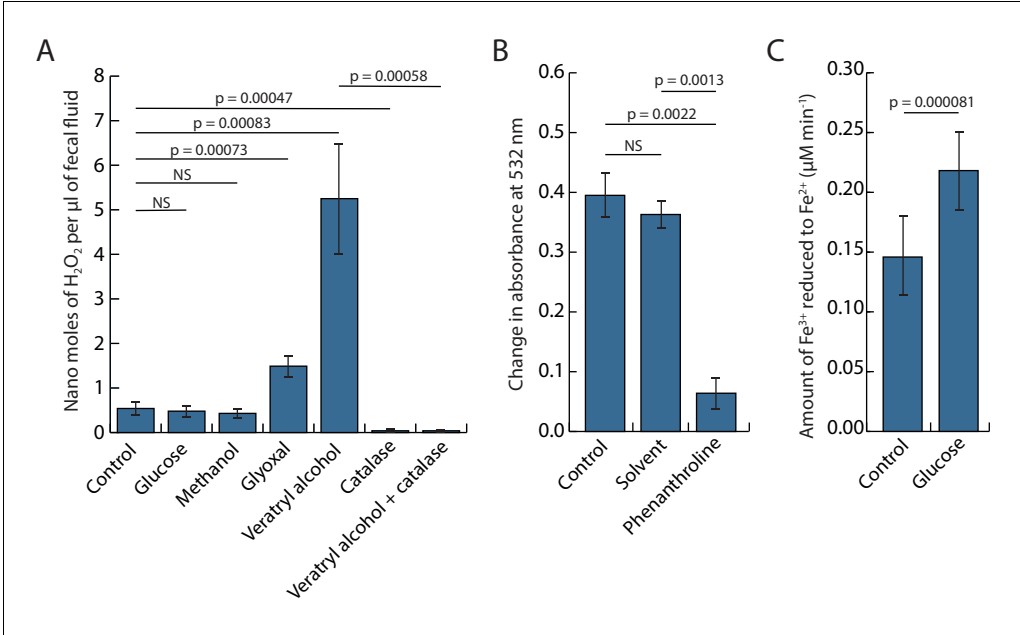

**Figure 4.** Bioassays to demonstrate that inorganic. Fenton chemistry must be taking place when ant fecal fluid is exposed to oxygen while being deposited on leaf pulp pellets chewed by the ants, testing the key interactions hypothesized in *Figure 2*. (A) Bar plot showing the concentrations (means ± SE across six colonies) of hydrogen peroxide in fecal fluid after adding potential substrates (glucose, methanol, glyoxal, or veratryl alcohol) of glucose-methanol-choline (GMC) oxidoreductases with or without the hydrogen peroxide degrading enzyme catalase. One-way ANOVA showed a highly significant overall effect of treatments: $F_{6,36}$ = 14.597, p=2.863e-08, and pairwise post hoc t-tests on matching samples from the same ant colonies (corrected for multiple testing with the Holm–Bonferroni method) confirmed both the enhancing effects of veratryl alcohol and glyoxal and the inhibiting effect of catalase (p-values in plot unless non-significant) (NS). (B) Deoxyribose assays showing that ant fecal fluid has the capacity to produce hydroxyl radicals (means ± SE across six colonies). Phenanthroline is known to work as an iron chelator and significantly reduced degradation of 2-deoxy-D-ribose while the solvent (methanol) of phenanthroline did not. Paired t-tests followed the same protocol as in the **A**-panel except that the overall ANOVA was omitted because there were only three means to compare. (C) A Ferrozine assay (means ± SE across six colonies) showing the capacity of ant fecal fluid to reduce $Fe^{3+}$ to $Fe^{2+}$, confirming that addition of glucose increases the rate of iron reduction. Statistics as in the **B**-panel.

The online version of this article includes the following source data for figure 4:

**Source data 1.** Data used for *Figure 4A, B, and C*.

from different ant colonies, and for many of them we were able to measure the corresponding enzymatic activity using biochemical assays (this study and *Schiøtt et al., 2010*; *De Fine Licht et al., 2013*; *Kooij et al., 2014b*; *Kooij et al., 2016*). These results strongly suggest that most if not all of the fecal proteins have condition-dependent adaptive significance, have been selected to avoid degradation in the digestive system of the ants, and serve the efficiency of the entire symbiosis between farming ants and fungal cultivars.

While many of the enzyme activities found in the ant fecal fluid have been discussed in previous publications, the roles of phosphate-liberating enzymes and lipidolytic enzymes have not. Just like nitrogen and carbon, phosphorus is a major macronutrient required for many biological processes. Enzymatic degradation of organic matter to liberate C, N, and P is known to take place in a stoichiometrically balanced way (*Sinsabaugh et al., 2009*), so it is not surprising that the fecal fluid also contains a range of enzymes able to release phosphorus from various substrates. That alkaline phosphatase (Protein ID 75 in *Table 1*) had the highest abundance of all fecal proteins underlines the importance of phosphorus mineralization. Several enzymes able to release phosphorus from phytin were also present. Phytin is a phosphorus-storage compound in plant seeds, but also occurs in pollen and vegetative tissues (*Brinch-Pedersen et al., 2002*). Laboratory colonies of *A. echinatior* were occasionally fed with dry rice grains, but in nature leaf-cutting ants are not known to forage on

seeds to any significant degree (*Kooij et al., 2014a*). However, they do harvest flowers, whose pollen may be a source of phytin. Finally, the fecal fluid contained ribonucleases that will degrade RNA, another major source of phosphorus. The fecal lipidolytic enzymes had low abundances overall. Apart from digesting lipids into assimilatory products, these enzymes may also hplay a function in degrading cell membranes to liberate cytosolic compounds as has been shown for biotrophic fungi (*Ghannoum, 2000*). They may also interfere with signal transduction pathways involved in plant resistance toward pathogens (*Wu and Li, 2015*), but all these putative functions require that the plant substrate is alive for a while after having been fragmented and placed in the fungus garden, which may well be the case.

## Multiple lines of evidence for fecal fluid Fenton chemistry after mixing with fresh leaf substrate

The key compounds hypothesized to be involved in inorganic Fenton chemistry (GMC oxidoreductases, hydrogen peroxide, high concentrations of iron, and glucose dehydrogenase) were always found in all four colonies for which we investigated fecal fluid samples. Compared to white-rot fungi, brown-rot fungi are believed to have a reduced array of enzymes targeting lignocellulose, and to specifically lack lignin peroxidases, which is also the case for *L. gongylophorus* (*Nygaard et al., 2016*). Brown-rot fungi compensate this lack of enzymatic potential by using Fenton chemistry to degrade lignocellulose while producing extracellular hydroxyl radicals (*Arantes and Goodell, 2014*). In the Fenton reaction, hydrogen peroxide reacts with ferrous iron ($Fe^{2+}$) and hydrogen ions to produce ferric iron ($Fe^{3+}$), water, and hydroxyl radicals (*Guillén et al., 2000*; *Arantes et al., 2012*). The very reactive hydroxyl radicals produced in these free-living fungi are known to damage a wide range of biomolecules (*Halliwell et al., 1988*). Their putative functions are: 1. to release soluble carbohydrate fragments from plant cell walls (*Schweikert et al., 2002*) and 2. to reduce the viscosity of solutions containing plant cell wall components such as xyloglucan, pectin, and cellulose (*Fry, 1998*), as expected when degradation of these polysaccharides has taken place. The small size of hydroxyl radical molecules allows them to initiate penetration and cleavage of lignin, cellulose, and hemicellulose polymers in preparation for further degradation into assimilatory monomers by specific enzymes (*Arantes et al., 2012*). Brown-rot fungi are believed to use GMC oxidoreductases or copper radical oxidases to aerobically oxidize fungal alcohols or aldehydes in order to produce hydrogen peroxide. In this process they then obtain the required iron from the wood substrate (*Arantes et al., 2012*) or via actively mycelial transport into the degrading wood (*Kirker et al., 2017*). It has been suggested that fungal laccases are then oxidizing hydroquinones into the semiquinones that react with ferric iron ($Fe^{3+}$) to produce quinones and ferrous iron ($Fe^{2+}$), which would produce the recycling process needed to maintain the Fenton reaction over time (*Guillén et al., 2000*).

It is intriguing that the details of Fenton chemistry in brown-rot fungi have remained rather little known in spite of the economic importance of these fungi (*Arantes and Goodell, 2014*), and that the leaf-cutting ants offered an unexpected window for understanding this elusive process. The laccases, GMC oxidoreductases, and copper radical oxidases were similarly found in high quantities in the fecal fluid of *Acromyrmex* leaf-cutting ants, consistent with Fenton chemistry being a standard procedure in this fungus-farming symbiosis to degrade plant cell walls and/or toxic secondary plant polymers meant to deter herbivores (*Figure 2*). We could verify that the identified GMC oxidoreductases were aryl alcohol oxidases (*Figure 3*), which are known to produce hydrogen peroxide from aromatic alcohols. We also verified that fecal fluid contains concentrated hydrogen peroxide and that the concentration increased further after adding veratryl alcohol or glyoxal, which are substrates for GMC oxidoreductases and copper radical oxidases, respectively (*Guillén et al., 1992*; *Daou and Faulds, 2017*). We further documented that the iron content of the fecal fluid is maintained at levels that are amply permissive for Fenton chemistry to occur. Also the acid pH of ca. 4 in the fecal fluid should be ideal for Fenton chemistry (*Bishop et al., 1968*; *Erthal et al., 2004*) although we cannot be sure this is the ambient pH because the plant material might also influence the final pH in the substrate pellet. Finally, a separate assay showed that fecal fluid components degrade deoxyribose into thiobarbituric acid-active products, which can only be explained by the presence of hydroxyl radicals (*Halliwell et al., 1988*). To obtain more direct evidence for the produced hydroxyl radicals contributing to plant substrate breakdown, future research should focus on using Fourier transform infrared spectroscopy to analyze plant cell wall preparations subjected to fecal fluid degradation, a type of analytical chemistry that was beyond the scope of the present study. Without such validation,

we cannot rule out other possible functions of these radicals such as general sanitation of the plant substrate. However, such alternative function has never been conjectured or shown for the ant fecal fluid, whereas brown-rot fungi are widely believed to use hydroxyl radicals for substrate degradation (*Arantes and Goodell, 2014*). We also note that specialized functions of the ant fecal fluid aimed at plant substrate degradation are uncontroversial (*Boyd and Martin, 1975b*; *Boyd and Martin, 1975a*; *Martin and Martin, 1971*; *Kooij et al., 2014b*; *Rønhede et al., 2004*; *Schiøtt et al., 2010*; *De Fine Licht et al., 2013*; *Kooij et al., 2016*). Our Fenton chemistry scenario is therefore supported by consistent inferential evidence, but in need of further experimental validation.

The high concentrations of laccase LgLcc1 (Protein ID 97 in *Table 1*) in the ant fecal fluid suggest a similar mechanism for maintaining reliable conversion of $Fe^{3+}$ back to $Fe^{2+}$ (using hydroquinone as an electron donor) as the Fenton chemistry known from brown-rot fungi (*Guillén et al., 2000*). However, the glucose dehydrogenase (Protein ID 145 in *Table 1*) encoded by the ant genome was also abundant in the fecal proteome and could convert $Fe^{3+}$ back to $Fe^{2+}$ via simultaneous oxidization of glucose. We verified this putative function as likely with a Ferrozine-based assay (*Goodell et al., 2006*), including its dependence on glucose availability. Interestingly, however, preliminary experiments indicated that adding laccase substrates did not influence the velocity of this reaction, which suggests that glucose liberated from the plant polysaccharides is primarily used to keep the Fenton reaction going. This substrate dependence may provide an additional feedback loop for the farming symbiotic partnership to control the inorganic Fenton chemistry process. The ant-derived glucose dehydrogenase (Protein ID 145 in *Table 1*) alternative might allow the ants to also use the fungal laccases specifically for the decomposition of secondary plant compounds as found in one of our previous studies (*De Fine Licht et al., 2013*). That study also suggested that LgLcc1 (Protein ID 97 in *Table 1*) came under positive selection in the common ancestor of the leaf-cutting ants, which would be consistent with laccase becoming redundant for iron reduction and therefore undergoing directional selection for other functions.

## Symbiotic complementarity is necessary for making fast Fenton chemistry work

The assembly of complementary components from different sources implies that the fungal cultivar and the farming ants synergistically cooperate to accomplish the continued production of hydroxyl radicals, but only as long as Fenton chemistry is needed. The ants do not appear to keep adding fecal fluid to existing Fenton pellets and they take them apart when they have turned black after some hours to distribute the Fenton exposed leaf fragments over the actively growing ridges of the fungus garden where fungal decomposition takes over. Both the processing efficiency and the ultimate behavioral control by the ants strongly suggest refined co-adaptation at the molecular level to enable colonies to operate as integrated functional herbivores. An earlier study found a high expression of an NADH-quinone oxidoreductase in fungus gardens of *A. echinatior* (*Grell et al., 2013*), an enzyme converting quinones to hydroquinones that can react with $Fe^{3+}$ to form $Fe^{2+}$ after conversion to semiquinones by laccase. This is consistent with the ant encoded glucose dehydrogenase being relevant only for the degradation process in the Fenton pellets, and with NADH-quinone oxidoreductases taking over after the leaf fragments have been dispersed over the garden and the fungal hyphae decompose what remains of the substrate.

Aryl alcohol oxidases use aromatic alcohols as a substrate to produce hydrogen peroxide. White-rot fungi produce aromatic veratryl alcohol as a secondary metabolite, which plays several important roles for ligninolysis (*Jensen et al., 1994*). Veratryl alcohol is derived from phenylalanine via the phenylalanine/tyrosine pathway that has previously been shown to be upregulated in the gongylidia of the *A. echinatior* cultivar *L. gongylophorus* (*De Fine Licht et al., 2014*). This might suggest that veratryl alcohol (or a derivative thereof) is produced in the gongylidia of the fungal cultivar to be ingested but not digested by the ants so it can function in the fecal fluid to facilitate the initial stages of substrate degradation. The hydrogen peroxide needed in the Fenton reaction may also be derived from the activity of the glyoxal oxidases that we identified, which use various aldehydes as substrate for hydrogen peroxide production (*Daou and Faulds, 2017*). Such aldehydes could then either originate from the oxidized aryl alcohols produced by the aryl alcohol oxidases, or could be direct degradation products from plant cell wall components (*Daou and Faulds, 2017*). These multiple parallel pathways suggest that the co-evolved interactions between inorganic Fenton chemistry

and organic enzyme functions are likely to have robust operational functionality under a variety of environmental (i.e. substrate) conditions.

Finally, as we explained in the introduction, it is important to note that hydroxyl radicals are highly toxic also for the organisms producing them. This probably constrains the use of naturally produced hydroxyl radicals for biomass degradation unless the Fenton chemistry producing these radicals remains distal to hyphal growth and is terminated by the time hyphae grow into the substrate patches pre-treated in this manner. This complication will slow down the rate of decomposition in free-living fungi, but the leaf-cutting ant symbiotic partnership appears to have resolved this constraint by compartmentalizing the complementary elements of initial aggressive break down of plant substrate. Although the necessary compounds for the Fenton reaction are all present in the guts of the ants, the low oxygen pressure there would presumably prevent the production of hydrogen peroxide inside the ants and thereby any collateral damage to ant tissues. Defecation onto concentrated pellets of chewed leaf fragments thus ensures that the reaction does not start until atmospheric oxygen is available. Consequently, the Fenton pellets appear to function as small bioconversion reactors that are spatially isolated from both the fungal and the ant tissues so that potentially harmful side-effects are avoided. Also in this respect robustness against potential malfunction appears to have evolved. Our recent genome sequencing study of *Acromyrmex* ants and their cultivars revealed that *A. echinatior* has more glutathione S-transferase genes than other (non-attine) ants, despite having fewer detoxification genes in general (*Nygaard et al., 2011*). Glutathione S-transferase is known to be involved in the breakdown of oxidized lipids, which would likely be produced if reactive oxygen species would be formed in the gut and react with the lipid cell membranes of the gut tissues. This suggests that physiological mechanisms to ameliorate inadvertent damage caused by hydroxyl radicals may be in place, should they appear in the ant gut before the excretion of fecal fluid.

## Implications for our general understanding of the leaf-cutting ant symbiosis and its bioconversion efficiency

The extent to which cellulose is degraded in the fungus garden has been widely disputed (see *Grell et al., 2013* and references therein). It is noteworthy that the fecal fluid proteome only contained a single 1,4-beta-glucanase (Protein ID 55 in *Table 1*) to target the cellulose backbone, and that this protein was found in low quantities and in only three of the four examined colonies. We also found a single 1,4-beta-glucosidase (Protein ID 73 in *Table 1*) that releases glucose moieties from cellobiose and oligosaccharides, but this enzyme is also targeting hemicellulose in addition to cellulose. Finally, effective degradation of cellulose normally requires the enzyme cellobiohydrolase, which was only found in one colony in very low quantities (*Supplementary file 1*). These findings are consistent with our earlier inference (*Grell et al., 2013*; *Moller et al., 2011*) that cellulose degradation is not a primary function of the fecal fluid, supporting earlier findings that recalcitrant plant cell wall cellulose is not degraded to a significant degree in *Acromyrmex* fungus gardens, but discarded from the colony as waste (*Moller et al., 2011*). Our finding that Fenton chemistry may be used for breaking chemical bonds in cell wall lignocellulose and is likely sufficient to give hyphal degradation enzymes access to their target substrates in the interior of plant cells, appears to corroborate this notion. More complete cellulose degradation would release an excess of assimilatory sugars for the fungal symbiont but without simultaneously providing nitrogen or phosphorus, which are the limiting factors for growth in any functional herbivore (*Elser et al., 2007*). Such sugars could well be a burden for the symbiosis if they would allow inadvertent microorganisms to thrive in fungus gardens (*Grell et al., 2013*).

While the documentation of Fenton chemistry substantially increases our understanding of the way in which *Acromyrmex* leaf-cutting ants could become crop pests by robust functional herbivory, it is intriguing that we have never observed Fenton pellets in any of the *Atta* colonies that we have maintained in the lab for ca. 25 years. This suggests that the colonies of this other genus of leaf-cutting ants, which are approximately two orders of magnitude larger in size and considerably more damaging as agricultural pests have evolved alternative mechanisms to boost the efficiency and robustness of their functional herbivory. Given that also *Atta* colonies produce enormous amounts of cellulose rich waste, we do not expect that the explanation for *Atta*'s success as functional herbivores will be fundamentally different. This is because both genera rear very closely related lineages of the *L. gongylophorus* cultivars in sympatry at our Panamanian sampling site (*Kooij et al., 2015b*)

and across their Latin American distributions (*Silva-Pinhati et al., 2004*). We suspect therefore that obtaining a fecal fluid proteome of *Atta* might shed interesting light on whether these ants can also use forms of inorganic chemistry, or whether they rely on an enzymatic innovation not available in *Acromyrmex*. Another point of interest is that human in vitro experiments with Fenton chemistry have produced up to fivefold increases in the production of sugars from feedstock, although the efficiency remains highly dependent on the plant material used (*Kato et al., 2014*; *Jung et al., 2015*; *Hegnar et al., 2019*). A major challenge in these human bioconversion applications has been that the Fenton generated reactive oxygen species degrade the very enzymes that are instrumental in maintaining sustainable conversion efficiency. An interesting venue for further research is therefore to elucidate how *Acromyrmex* fecal fluid degradation enzymes avoid being harmed by the hydroxyl radicals that accompany them.

Finally, we are used to think of our own crops as being in continuous need of manure to boost nutrient availability and growth. The term manure now seems inappropriate for attine ant fecal fluid even though deposition of fecal droplets in the fungus garden seems to suggests this analogy. In retrospect, abandoning this term seems logical, because the ants rear a heterotrophic crop and it seems hard to imagine how fresh leaf substrate can be further enriched with nutrients. Our present results suggest that it is also ambiguous to use the term 'fecal' for the droplets that leave the hindguts of leaf-cutting ants. In a structural sense this is fecal material, but in a functional sense this fluid now seems to be analogous to a joint symbiotic circulation system tinkered together by natural selection over millions of years. The primary function of this higher-order circulation system appears to be optimizing the distribution of gongylidia-produced decomposition compounds to maintain the combined superorganismal body of ants, ant brood, and fungus garden. In this perspective, it also becomes ambiguous which partner is the farmer or the crop, because the relationship is symmetrical in its mutual dependence and the cultivar has remarkable agency even though it neither has brains to coordinate or legs to move around. This perspective is worked out in more detail elsewhere (*Boomsma, 2022*) and suggests that we need to be careful in applying anthropomorphic language to naturally evolved symbioses outside the human domain.

## Materials and methods

### Biological material

Colonies of *Acromyrmex echinatior* (numbers Ae150, Ae160, Ae263, Ae322, Ae356, Ae372, Ae480, and Ae490) were collected in Gamboa, Panama between 2001 and 2010 and maintained in the laboratory at ca. 25°C and ca. 70% relative humidity (*Bot and Boomsma, 1996*) where they were supplied with a diet of bramble leaves, occasional dry rice, and pieces of apple. Fecal droplets were collected by squeezing large worker ants with forceps on the head and abdomen until they deposited a drop of fecal fluid. For mass spectrometry, 0.5 µl of double distilled and autoclaved water was added to the droplet before it was collected with a micro pipette and transferred to an Eppendorf tube with Run Blue loading buffer (Expedeon Inc). For the iron assay, Fenton reaction assay, hydrogen peroxide assay, and iron reduction assay, the fecal droplets were deposited on parafilm and collected undiluted with a 5 µl glass capillary.

### Mass spectrometry

Ten fecal droplets from each of the four colonies were run on a 12% Run Blue polyacrylamide gel (Expedeon Inc) in a Mini-Protean II electrophoresis system (Biorad) for about 10 min at 90 mA. The gel was stained in Instant Blue Coomassie (Expedeon Inc) overnight at room temperature. The stained part of each lane was then cut out with a scalpel and transferred to Eppendorf tubes, after which the samples were sent to Alphalyse (Denmark) for analysis.

The protein samples were reduced and alkylated with iodoacetamide (i.e. carbamidomethylated) and subsequently digested with trypsin. The resulting peptides were concentrated by Speed Vac lyophilization and redissolved for injection on a Dionex nano-LC system and MS/MS analysis on a Bruker Maxis Impact QTOF instrument. The obtained MS/MS spectra were analyzed using the Max-Quant software package version 1.5.4.1 (*Cox and Mann, 2008*) integrated with the Andromeda search engine (*Cox et al., 2011*) using standard parameters. The spectra were matched to a reference database of the predicted *Acromyrmex echinatior* proteome (*Nygaard et al., 2011*) combined

with published predicted *Leucocoprinus gongylophorus* proteomes originating from the *Acromyrmex echinatior*, *Atta cephalotes*, and *Atta colombica* fungal symbionts (*Aylward et al., 2013*; *Nygaard et al., 2011*; *Nygaard et al., 2016*), as well as single protein sequences predicted from manually PCR amplified and sequenced cDNA sequences (*Schiøtt et al., 2010*; *De Fine Licht et al., 2013*; *Kooij et al., 2016*; *Kooij et al., 2014b*; *Schiøtt et al., 2008*). As these published fungal proteomes contained many redundant protein sequences, a non-redundant reference protein database was constructed using CD-Hit (RRID:SCR_007105) (*Li and Godzik, 2006*) keeping only the longest sequences of the sets of 100% matching sequences. Annotation of the identified protein sequences was performed using Blast2GO (RRID:SCR_005828) (*Hegnar et al., 2019*) combined with manual BlastP searches against the NCBI nr database.

## Iron level measurements

Fecal fluid iron content was measured according to *Fish, 1988*. Fecal fluid from 20 workers from each of six colonies (Ae150, Ae160, Ae263, Ae356, Ae480, and Ae490) was collected with glass capillaries and weighed in 0.2 ml PCR tubes to determine the volume of each sample, assuming a density of 1 g/ml. The samples were diluted to 100 µl with 0.1 M HCl, mixed with Reagent A (0.142 M $KMnO_4$, 0.6 M HCl), and incubated for 2 hr at 60˚C. Subsequently 10 µl Reagent B (6.5 mM Ferrozine, 13.1 mM neocuproine, 2 M ascorbic acid, 5 M ammonium acetate) was added to each sample and incubated for 30 min at room temperature, after which 150 µl of this mixture was transferred to a microtiter plate and the absorbance at 562 nm measured in an Epoch microplate spectrophotometer (BioTek). Samples with known amounts of iron were processed in parallel with the unknown samples to produce a standard curve from which the iron content of the unknown samples could be determined. Three series of standards were made from ferrous ethylenediammonium sulfate tetrahydrate, ferrous sulfate heptahydrate, and ferric chloride hexahydrate. In all cases a solution of 5 mg/l iron in 0.1 M HCl was used to make a dilution series ranging from 78.1 µg/l to 5 mg/l. In addition, a blank sample without iron was included and used to correct for background absorbance.

## Hydroxyl radical measurements

0.5 µl fecal fluid from each of the colonies Ae150, Ae160, Ae263, Ae356, Ae480, and Ae490 was diluted with 0.5 µl distilled $H_2O$ and mixed with either 1 µl $H_2O$, 1 µl 40 mM 1,10-phenanthroline (dissolved in methanol to 200 mM and diluted 1:4 with distilled $H_2O$) or 1 µl 20% methanol. After 15 min incubation at room temperature, 2 µl substrate (100 mM 2-deoxy-D-ribose in 0.1 M sodium acetate buffer pH 5.0) was added, and samples were incubated for 30 min at room temperature. Then, 4 µl of a solution of 1% 2-thiobarbituric acid in 50 mM NaOH was added followed by addition of 4 µl of 2.8% trichloroacetic acid. The samples were then incubated for 20 min at 99.9˚C in a PCR machine, after which 2 µl was used for measuring the absorbance at 532 nm using a NanoDrop ND-1000 spectrophotometer (*Halliwell et al., 1988*). Control samples without addition of 2-deoxy-D-ribose were made in parallel and the absorbance value subtracted from the absorbance values of the samples. Similarly, control samples with 2-deoxy-D-ribose added but without fecal droplets were made in parallel so also their absorbance value could be subtracted from those of the samples. Control samples without both 2-deoxy-D-ribose and fecal droplets were also run in parallel and these absorbance values were added to the final values to compensate for subtracting the absorbance values of two controls from each sample value.

## $H_2O_2$ measurements

0.5 µl fecal fluid from each of the colonies Ae150, Ae160, Ae263, Ae356, Ae480, and Ae490 was mixed with either 150 µl demineralized $H_2O$ or with 150 µl 10 mM glucose, 10 mM glyoxal, 10 mM veratryl alcohol, 10 mM methanol, or 100 U catalase. The samples were incubated for 60 min at 30˚C in a PCR machine. To avoid direct oxidation of the amplex red substrate by laccases in the fecal fluid, the samples were transferred to Amicon Ultra-0.5 Centrifugal filters with 3 Kda membranes (Merck Millipore UFC500396) and centrifuged at 14,000 g for 30 min. 100 µl flow through was transferred to a microtiter plate and 100 µl amplex red reagent (100 µM Ampliflu red, 2 U/ml horse radish peroxidase, in 0.1 M phosphate buffer pH 7.4) (*Zhou et al., 1997*) was added, after which the absorbance at 560 nm was measured for 30 min using an Epoch microplate spectrophotometer (BioTek). This absorbance measure was then used to calculate the $H_2O_2$ concentration using a standard curve

made from a twofold serial dilution of $H_2O_2$ ranging from 1.56 µM to 100 µM and including a blank sample without $H_2O_2$. The standard samples were assayed in parallel with the fecal fluid samples.

## Iron reduction measurements

One microliter fecal fluid from each of the colonies Ae150, Ae160, Ae263, Ae356, Ae480, and Ae490 was mixed with 71 µl demineralized $H_2O$, 10 µl 1 mM $FeCl_3 \cdot 6 H_2O$, 115 µl 0.1 M acetate buffer pH 4.4, and either 23 µl $H_2O$ or 23 µl 100 mM glucose. Subsequently, 10 µl 1% FerroZine was added and the absorbance at 562 nM was measured during a 30 min period using an Epoch micro-plate spectrophotometer (BioTek) (*Goodell et al., 2006*). The rates of change in absorbance from 5 to 30 min were used to calculate the amount of $Fe^{3+}$ reduced to $Fe^{2+}$ per minute, using a standard curve made from a twofold serial dilution of $FeSO_4 \cdot NH_3CH_2CH_2NH_3SO_4 \cdot 4 H_2O$, ranging from 5 mM to 0.078 mM and including a blank sample without iron. The standard samples were assayed in parallel with the fecal fluid samples.

## Statistical data analyses

All analyses were done in RStudio Version 0.99.491 (RRID:SCR_000432). Shapiro–Wilk tests were used to test whether differences in response variables between treatments within the same ant colonies followed a normal distribution. For hydrogen peroxide measurements a square root transformation of response data was applied to make the data pass the Shapiro–Wilk test. Planned comparisons between treatments were performed with paired t-tests (by pairing measurements of samples coming from the same ant colonies). p-values were adjusted for multiple comparisons using the Holm–Bonferroni method.

## Phylogenetic analyses

The five identified fecal fluid GMC oxidoreductase sequences were aligned with 15 basidiomycete GMC oxidoreductase sequences belonging to each of the five functional GMC oxidoreductase protein groups presented in *Ferreira et al., 2015*, using M-coffee (http://tcoffee.crg.cat/apps/tcoffee/do:mcoffee) with default parameter settings. The alignment file was then used in PhyML (RRID:SCR_014629) with Smart Model Selection (http://www.atgc-montpellier.fr/phyml-sms/) and aLRT SH-like support values. The best DNA substitution model turned out to be LG+G+I+F.

## Acknowledgements

We thank the Smithsonian Tropical Research Institute (STRI), Panama, for providing logistic help and facilities during our field work in Gamboa.

## Additional information

### Funding

| Funder | Grant reference number | Author |
| --- | --- | --- |
| Danmarks Grundforsknings-fond | DNRF57 | Jacobus J Boomsma |
| H2020 European Research Council | Advanced Grant 323085 | Jacobus J Boomsma |

The funders had no role in study design, data collection and interpretation, or the decision to submit the work for publication.

### Author contributions

Morten Schiøtt, Conceptualization, Formal analysis, Investigation, Methodology, Writing - original draft, Writing - review and editing; Jacobus J Boomsma, Conceptualization, Funding acquisition, Writing - original draft, Project administration, Writing - review and editing

Author ORCIDs
Morten Schiøtt https://orcid.org/0000-0002-4309-8090
Jacobus J Boomsma https://orcid.org/0000-0002-3598-1609

Decision letter and Author response
Decision letter https://doi.org/10.7554/eLife.61816.sa1
Author response https://doi.org/10.7554/eLife.61816.sa2

## Additional files

### Supplementary files
• Supplementary file 1. Data for all fecal fluid proteins identified in the study.

• Transparent reporting form

### Data availability

The mass spectrometry data generated in this project as well as the amino acid sequences used for protein identification have been deposited to the ProteomeXchange Consortium via the PRIDE (Perez-Riverol et al. (2019). Nucleic Acids Research 47:D442–D450.) partner repository with the dataset identifier PXD016395 (http://www.ebi.ac.uk/pride/archive/projects/PXD016395). All other data are provided in the Supplementary file S1 and the source data files provided for Figures 3 and 4.

The following dataset was generated:

| Author(s) | Year | Dataset title | Dataset URL | Database and Identifier |
|---|---|---|---|---|
| Schiøtt M, Boomsma JJ | 2020 | Leaf-cutting ant feces LC-MS/MS | http://www.ebi.ac.uk/pride/archive/projects/PXD016395 | PRIDE, PXD016395 |

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
