## [Decision Letter]

**Acceptance summary:**

The study by Schiøtt and Boomsma moves our understanding of the intricate mutualistic interaction between leaf-cutter ants and their fungus a significant step forward. Using a combination of proteomics and subsequent specific chemical tests to understand the nature of leaf degradation the authors present compelling evidence that a Fenton reaction plays a crucial role in the degradation of the plant material, which is otherwise hard to break down. The chemicals for the Fenton reaction are provided in a unique way by both the fungus and the ants, stressing once more the very tight mutualism between these species.

**Decision letter after peer review:**

Thank you for submitting your work entitled "Proteomics reveals synergy between biomass degrading enzymes and inorganic Fenton chemistry in leaf-cutting ant colonies" for consideration by *eLife*. Your article has been reviewed by four peer reviewers, one of whom is a member of our Board of Reviewing Editors, and the evaluation has been overseen by Meredith Schuman as the Senior Editor.

Four reviewers have read this study and they are all enthusiastic about the results. All reviewers made a number of comments that should be addressed before publications.

The most prominent suggestions are:

1) A recurrent issue in several comments raised by the reviewers is a lack of clarity. Given the interdisciplinary nature of the study, particular care should be given to make the material better accessible for non-specialists. This requires a better organisation of the way certain aspects of the material is presented (including the logical steps from proteome analysis), explanation of terminology and better explanation of aspects that are specific to the chemical details of the finding.

2) All reviewers had issues with specific aspects of the description of the chemical aspects of the study. Here consistency and clarity is of particular importance.

3) The title should be more descriptive with regard to the study.

4) The study refers at various places to brown rot fungi. The relevance of this, the physiological, biochemical and phylogenetic link to the ant-fungus is not always clear.

All the comments by the four reviewers are listed below.

Reviewer #1:

The study by Schiøtt and Boomsma moves our understanding of the intricate mutualistic interaction between leaf-cutter ants and their fungus a significant step forward. Using a combination of proteomics and subsequent specific chemical tests to understand the nature of leaf degradation the authors present compelling evidence that a Fenton reaction plays a crucial role in the degradation of the plant material, which is otherwise hard to break down. The chemicals for the Fenton reaction are provided in a unique way by both the fungus and the ants, stressing once more the very tight mutualism between these species.

I am not good enough in chemistry to judge if the chemical tests presented are as clear in their interpretation as it is presented here; this needs to be assessed by a biochemist. Other than this I have little comments.

My main comment is to make the study more accessible to non-specialist readers. The text meanders at places and needs more of a clear line. Terminology needs to be explained better. I had to consult Wikipedia several times while reading this study.

The title is not very clear. As it stands it misses, what I consider a main point, namely that biomass degradation is achieved by a synergy of fungal and ant chemistry.

Reviewer #2:

The authors have conducted a thorough study of the fecal fluid proteome in *Acromyrmex echinator* and its potential adaptive function in maintaining the ants' symbiosis with the fungus *Leucocoprinus gongylophorus*. The authors provide strong evidence that Fenton reactions occur within pellets of chewed plant matter mixed with ant fecal fluid, and they suggest these reactions enable the efficient digestion of plant material. Weaker evidence suggests both ant and fungal proteins maintain the Fenton reactions. Overall, this study advances our understanding of chemical mechanisms permitting a textbook example of symbiosis, and it raises intriguing questions for future study regarding the evolutionary origins of this and other leaf-cutter ant – fungal symbioses.

1) Several points regarding glucose dehydrogenase and its proposed role in the Fenton reaction need to be clarified

– How did the researchers choose to focus on this particular protein as the candidate for the reducing agent in the Fenton reaction? Why did this protein stand out from the list (besides being the most abundant of the ant proteins)? Does it play a similar role in other systems?

– In the Discussion, the reader learns that laccases are apparently an alternative to glucose dehydrogenase in a Fenton reaction, and these laccases are also present in the fecal fluid, so this raises even more questions as to why glucose dehydrogenase was chosen as the candidate.

The story with the laccases, including the preliminary experiments, should be introduced sooner (e.g. when introducing the hypothesis involving glucose dehydrogenase)

– What is the relationship between "glucose dehydrogenase" and "GMC oxidoreductase" (from the protein table, it seems that glucose dehydrogenase is a type of GMC oxidoreductase, but this is not clear in the text). Also, what is its relationship to aryl alcohol oxidases, and is it also present in brown-rot fungi?

– Is glucose dehydrogenase the only enzyme that would increase its activity in response to additional glucose? (i.e. could the effect of adding glucose be caused by another candidate enzyme?)

2) It is not clear how long the Fenton reactions can be maintained, given the iron recycling which was demonstrated. The authors imply that Fe2+ is not the limiting reagent, but the evidence for this is not apparent. Iron reduction was measured at "considerable capacity", but in relation to what? The comparison to human blood (Results and Discussion) is not relevant without additional context: How does it compare to the wood substrate of brown-rot fungi? How much is needed to maintain the Fenton reaction? How does the recycling rate compare to the consumption rate?

3) The authors argue that the fecal fluid of *A. echinator* contains a "specialized" set of proteins that have been adaptively selected to escape digestion. The main evidence they give is that the proteins consistently produce the same banding pattern on gels, and that most of the proteins were found in three of four colony samples. However, the reader is given no phylogenetic context for this comparison: do other ant species (or any other species) have fecal proteomes that appear more "random" on gels? Moreover, we know these ants always eat the same food, so why should we expect their fecal proteomes to vary over time or between colonies?

4) Language: Many sentences are quite long and complex. I recommend keeping sentences under 30 words as a rule of thumb. Also, there are several errors with grammar and punctuation, as well as misplaced modifiers (e.g. Introduction, Results), which make the paper laborious to follow. I recommend consulting with a professional editor to improve language use and overall clarity of the manuscript.

Reviewer #3:

Already decades ago it was suggested that the fecal fluid of leaf-cutter ants contains enzymes required for the breakdown of plant material that are produced by the fungal cultivar. While this was corroborated in recent studies on the proteome of the fecal fluid as well as genome evolution of both, the ants and the fungal cultivar, the mechanism of the actual breakdown of the plant material still remained a black box.

Based on a more in-depth proteome study of the fecal fluid of the leaf-cutter ant *Acromyrmex echinator* the authors can show that the fecal fluid predominantly contains fungal-produced enzymes. While fungal laccases were already known from the fecal fluids, here the authors present data that many enzymes may contribute to breakdown of plant material. Based on the idea that abundantly occurring oxidoreduxctases produce hydrogen peroxide the authors test the hypothesis that this molecule may react with iron to produce reactive oxygen radicals, i.e. the inorganic Fenton reaction would proceed after ants place a droplet of fecal fluid on ant-produced pellets of munched leaf material that are placed onto the surface of the fungal cultivar (“Fenton ball”). The reactive oxygen species may then damage plant cell walls to an extent that enables the fungal cultivar to enter into the cells and obtain nutrients.

Taken together I think that the experimental evidence showing that the Fenton reaction proceeds in the “Fenton balls” on the surface of the fungal cultivar is very strong. Overall, the study is of very high quality and is well worth publishing.

1) I suggest that the authors explain whether there is a link between brown rot fungi and the ant cultivar. Brown rot fungi typically use hydrogen peroxide and oxygen radicals to aid in breakdown of cellulose and hemicellulose. Is there a phylogenetic relationship of the ant cultivar to brown rot fungi? Or has the trait of producing H_2_O_2_ and the entailing reactions evolved multiple times?

2) In brown rot fungi aromatic alcohols play an important role. These alcohols are oxidized in order to produce H_2_O_2_. The aromatic alcohol veratryl alcohol produced by brown rot fungi was also used here in experimental procedures, where the authors showed that H_2_O_2_ is produced in larger quantities when it is present. Is anything known about the concentration of such aromatic alcohols in the fecal fluid of leaf-cutter ants? How would it be produced (and by whom?)?

3) Is there any evidence for a high concentration of citrate in the gut of leaf-cutter ants? Indirect evidence is presented by the authors (Results) with a Mollicute bacterium that has a citrate catabolizing pathway. However, is this bacterium part of the core microbiome? Any possibility to provide more direct evidence here? The Mollicute bacterium would only be an opportunist scavenging the citrate? In Sapountzis et al., 2018, the information is given that the leaf-cutter workers would consume relatively high levels of citrate with juices from freshly cut plant material. I think that this link may be more convincing.

4) I really like the point of the Discussion where the authors point out that the Fenton balls are spatially separated bioconversion reactors that prevent tissue damage of either ant of fungus when high levels of reactive oxygen is presumably produced.

5) A very interesting point that the authors touch for future studies is that other higher leaf-cutter ants of genus Atta do not seem to make use of mechanism for the breakdown of plant material studied here for *Acromyrmex*. Is this really a total black box? Or could the authors at least provide an idea how it works in Atta?

6) In contrast, I think the last section of the Discussion is only partially helpful. While I agree with the authors that the high level of physiological complementarity and the large proportion of fungal enzymes in the fecal fluid renders the term “manure” inappropriate I do think that the suggested term “bloodstream” is not very helpful. The superorganism does not have a bloodstream (and also not a joint symbiotic one). I am not sure whether this highly anthropomorphic analogy is helpful in the future. I suggest to tone down this paragraph and stress the functional physiological complementarity more.

Reviewer #4:

The work presented by Schiøtt and Boomsma is a very interesting study investigating how the Fenton reaction assists enzymatic digestion of plant material collected by leaf-cutting ants in fecal substrate pellets.

The text is very enjoyable to read.

The methodologies are well-described and the findings are of interest.

1) In the Introduction, it is stated that saprotrophs are not well adapted to efficiently degrade fresh plant material. I think there are many sparotrophic fungi well-adapted to do this.

2) Throughout the work, oxidoreductases have been categorized separately from other CAZY enzymes. For example, in the Results and in Table 1 and Figure 1. However, enzymes such as laccase (AA1), GMC oxidoreductase/aryl-alcohol oxidase (AA3), etc. are also classified as CAZY enzymes, although as having auxiliary activities. Could oxidoreductases be structured as a sub-category of CAZY enzymes to distinguish them from hydrolytic enzymes?

3) In the Results and Discussion, it is stated that brown rot fungi rely on iron already present in the wood to employ the Fenton reaction. This idea may be somewhat outdated as XRF studies in *Serpula lacrymans* have shown that this brown rot fungus actively increases the iron concentrations in wood before decomposition. See Kirker et al., 2017. Synchrotron-based X-ray fluorescence microscopy enables multiscale spatial visualization of ions involved in fungal lignocellulose deconstruction. Scientific Reports 7, Article number: 41798.

4) Discussion. I do not see how laccases in the fecal fluid can be considered functional analogs of brown rot recycling mechanisms of Fe(III) to Fe(II) directly. Laccases in the system described by Guillen et al., 2000 oxidise quinones. Reduced quinones in that system are responsible for reducing Fe(III) to Fe(II) and the production of H_2_O_2_. This sentence reads as if the laccases perform this function instead of the quinones.

5) Results. Not only enzymes can reduce Fe(III) to Fe(II) in brown rot fungi. The recycling of Fe(III) to Fe(II) in brown rot systems can be attributed to quinones as well.

---

## [Author Response]

Four reviewers have read this study and they are all enthusiastic about the results. All reviewers made a number of comments that should be addressed before publications.The most prominent suggestions are:1) A recurrent issue in several comments raised by the reviewers is a lack of clarity. Given the interdisciplinary nature of the study, particular care should be given to make the material better accessible for non-specialists. This requires a better organisation of the way certain aspects of the material is presented (including the logical steps from proteome analysis), explanation of terminology and better explanation of aspects that are specific to the chemical details of the finding.2) All reviewers had issues with specific aspects of the description of the chemical aspects of the study. Here consistency and clarity is of particular importance.3) The title should be more descriptive with regard to the study.4) The study refers at various places to brown rot fungi. The relevance of this, the physiological, biochemical and phylogenetic link to the ant-fungus is not always clear.

We have endeavoured to clarify along the lines summarized here, but could not find an alternative title that could capture more of the key focus of our paper and still remain within the maximally allowed number of characters. See below for a more detailed response.

All the comments by the four reviewers are listed below.Reviewer #1:[…] I am not good enough in chemistry to judge if the chemical tests presented are as clear in their interpretation as it is presented here; this needs to be assessed by a biochemist. Other than this I have little comments.My main comment is to make the study more accessible to non-specialist readers. The text meanders at places and needs more of a clear line. Terminology needs to be explained better. I had to consult Wikipedia several times while reading this study.The title is not very clear. As it stands it misses, what I consider a main point, namely that biomass degradation is achieved by a synergy of fungal and ant chemistry.

We struggled with the stringent limitation of maximally 120 characters being allowed for the title.

If we understand the reviewer well, (s)he finds that our present title:

“Proteomics reveals synergy in biomass conversion between fungal enzymes and inorganic Fenton chemistry in leaf-cutting ant colonies”

lacks the word “ant-chemistry”. However, all three keywords “synergy”, “fungal” and “ant” are there, so it is largely a matter of semantics whether one finds this title captures the key elements of our study. We have toyed with various alternatives but found none of them were improvements. This exercise also reminded us that we arrived at the present title on purpose because the chemistry participation of the ants is in fact rather minor and indirect compared to the more major and direct involvement of the fungal cultivar. The ant roles are of key significance but that is because of vectoring of fecal droplets and fine-tuned behavioral facilitation. Also the majority of the fecal fluid proteome is not ant derived (86 fungal/15 ant) and only a single enzyme involved in the Fenton reaction was encoded by the ant genome. We therefore concluded in the end that we prefer to keep the title unchanged because it will, together with the Abstract, convey the take-home messages of this study quite accurately.

The only alternative title that we could think of:

“Proteomics reveals that leaf-cutting ants and their fungal symbiont synergistically combine Fenton chemistry with enzymatic degradation of plant biomass”

exceeds the character limit, so the question is whether you feel this is a sufficient improvement to give dispensation from that rule.

Reviewer #2:[…] 1) Several points regarding glucose dehydrogenase and its proposed role in the Fenton reaction need to be clarifiedo How did the researchers choose to focus on this particular protein as the candidate for the reducing agent in the Fenton reaction? Why did this protein stand out from the list (besides being the most abundant of the ant proteins)? Does it play a similar role in other systems?

As mentioned in the text, we conjectured it there should be a mechanism to recycle oxidized iron into reduced iron in order to keep the Fenton reaction running. Scrutinizing Table 1 for proteins that could have such a function, the glucose dehydrogenase stood out because: 1) It is an oxidoreductase and therefore capable of mediating redox reactions. 2) Using glucose as an electron donor would mean that it would constantly be supplied with substrate because the degradation of plant material liberates glucose molecules in a constant flow. 3) It appeared to always be present in high quantities. We have not been able to find other examples in the literature that document the use of glucose dehydrogenase for this purpose, and we also do not claim to have proved beyond doubt that this is the prime function of the identified glucose dehydrogenase. But the fact that addition of glucose increased the iron reduction capability of the fecal fluid was consistent with this hypothesis, and glucose dehydrogenase has also been suggested to be used for glucose liberation functions in industrial Fenton chemistry, once more consistent with such a function being an obvious possibility. We note that glucose dehydrogenase may not directly catalyze the iron reduction reaction, but may depend on other enzymes catalyzing intermediate steps in the process, with the ultimate result of reducing oxidized iron. Finally, cellobiose dehydrogenase is known to be able to reduce oxidized iron with cellobiose as substrate, which adds to the plausibility of glucose dehydrogenase have a similar function.

– In the Discussion, the reader learns that laccases are apparently an alternative to glucose dehydrogenase in a Fenton reaction, and these laccases are also present in the fecal fluid, so this raises even more questions as to why glucose dehydrogenase was chosen as the candidate.

We explicitly reported in the manuscript that simple experiments adding laccase substrate to the fecal fluid did not affect its iron reducing capabilities, which suggests that laccases have roles that are independent of Fenton chemistry. We admit that more elaborate experiments on this would have been nice to do, but they are very time consuming and were thus beyond the scope of the present study.

So, while we do not rule out the possibility of laccase activity somehow interacting with Fenton chemistry, it does not seem an obvious possibility. We note that Fenton pellets are typically produced when colonies receive leaf-resources ad libitum, so it may be that laccase decomposition primarily works when resources are less abundant and glucose becomes a more valuable resource.

The story with the laccases, including the preliminary experiments, should be introduced sooner (e.g. when introducing the hypothesis involving glucose dehydrogenase)

A sentence on the preliminary laccase experiments has now been added to the Results section.

– What is the relationship between "glucose dehydrogenase" and "GMC oxidoreductase" (from the protein table, it seems that glucose dehydrogenase is a type of GMC oxidoreductase, but this is not clear in the text). Also, what is its relationship to aryl alcohol oxidases, and is it also present in brown-rot fungi?

It is correct that glucose dehydrogenase belongs to the GMC oxidoreductase protein family, but as it does not produce hydrogen peroxide, it cannot alone be mediating the Fenton reaction. Also, being an ant-encoded protein, it was not meaningful to include it into the gene tree in Figure 3, which had only fungal proteins. Glucose dehydrogenases are closely related to glucose oxidases (GOX), which means that they are also (but more distantly) related to aryl alcohol oxidases. They are present in both white rot fungi and brown rot fungi. As our manuscript was already quite long, we chose not to prioritize detailed further descriptions of these issues as that would have mostly distracted from the overall interpretational narrative. We have, however, now added the sentence: “Note that the ant encoded glucose dehydrogenase is also a GMC Oxidoreductase, but would not fit into this phylogeny of fungal proteins” to the legend of Figure 3 to make clear that we were aware of this issue.

– Is glucose dehydrogenase the only enzyme that would increase its activity in response to additional glucose? (i.e. could the effect of adding glucose be caused by another candidate enzyme?)

To our knowledge, glucose dehydrogenase is the only enzyme in the fecal fluid proteome that can use glucose as a substrate. However, there are a few hypothetical proteins in the list for which we do not know the function. Although our proteome is very complete, it also remains possible that some proteins somehow escaped being identified by the methods that we used.

2) It is not clear how long the Fenton reactions can be maintained, given the iron recycling which was demonstrated. The authors imply that Fe2+ is not the limiting reagent, but the evidence for this is not apparent.

We have not done experiments to look into this issue, partly because the fecal fluid material is only available in small quantities, which limits what kind of experiments can be feasibly done. In principle we believe the reaction can go on as long as oxygen, aromatic alcohols, and glucose are all present, of which the amount of alcohols will probably be the limiting component. However, as the fecal fluid also contains a lot of proteases, the limiting factor may in reality be proteolytic degradation of the relevant enzymes. To look into these issues, it will probably be needed to mix the various components in vitro at a much larger scale, which we have not tried yet. We have now deleted the sentence stating that the Fenton reaction appeared not to be limited by metal ions.

Iron reduction was measured at "considerable capacity", but in relation to what?

Unfortunately it is very difficult to compare this value to other studies, as we do not know the amount of the relevant enzymes in the fecal fluid. But what matters in this context is that we can conclude that iron reduction takes place in the fecal fluid and that the velocity of the reaction increases after adding glucose. The exact values are therefore not so important. We have now changed “considerable capacity” to “measurable capacity”.

The comparison to human blood (Results and Discussion) is not relevant without additional context: How does it compare to the wood substrate of brown-rot fungi?

We have now revised the relevant paragraphs, removing the comparison to human blood and instead focusing on how this value compares to other systems in which Fenton chemistry plays a role (Results and Discussion).

How much is needed to maintain the Fenton reaction? How does the recycling rate compare to the consumption rate?

As mentioned above, even though these kinds of experiments are relevant to do, the small amounts of fecal fluid material that can be obtained by pressing it out of individual workers and the possible interactions with many other substances in the fluid makes such experiments difficult without having an established in vitro model system. Also here, we do not claim to have solved all questions related to the proposed scenario, but present what we think is compelling cumulative evidence to suggest that Fenton chemistry plays a key role in the degradation of plant substrate in leaf-cutting ant fungus gardens. Thereby we will hopefully pave the way for further investigation of this subject in the same or other model systems.

3) The authors argue that the fecal fluid of A. echinator contains a "specialized" set of proteins that have been adaptively selected to escape digestion. The main evidence they give is that the proteins consistently produce the same banding pattern on gels, and that most of the proteins were found in three of four colony samples. However, the reader is given no phylogenetic context for this comparison: do other ant species (or any other species) have fecal proteomes that appear more "random" on gels? Moreover, we know these ants always eat the same food, so why should we expect their fecal proteomes to vary over time or between colonies?

First of all, proteins are very valuable sources of nitrogen, and most organisms would degrade and assimilate as much food protein as possible. That we repeatedly find the same set of proteins, and that these proteins are still functional after gut passage, and with functional roles relevant for degradation of plant biomass, cannot be a coincidence. It is true that the diet of leaf-cutting ants mostly consists of fungal hyphae, but all we say is that if the escape from digestion was just a coincidence, the banding pattern on the gels should be random, which it is definitely not. Consequently there must be a predictable reason why these proteins are not digested, and given that their putative functions make sense when interpreted as mediating a sequence of substrate degradation steps, it seems fair to interpret this phenomenon as an advanced symbiotic adaptation. We are not aware of studies looking into fecal proteomes of other ants, and studies on humans and other mammals typically find bacterial proteins and host proteins involved in inflammation. As we illustrated in the final Discussion paragraph, it appears that the fungus-farming symbiosis makes it ambiguous whether ant feces can still be considered as waste.

4) Language: Many sentences are quite long and complex. I recommend keeping sentences under 30 words as a rule of thumb. Also, there are several errors with grammar and punctuation, as well as misplaced modifiers (e.g. Introduction, Results), which make the paper laborious to follow. I recommend consulting with a professional editor to improve language use and overall clarity of the manuscript.

The manuscript has been revised to generally improve clarity of communication.

Reviewer #3:[…] 1) I suggest that the authors explain whether there is a link between brown rot fungi and the ant cultivar. Brown rot fungi typically use hydrogen peroxide and oxygen radicals to aid in breakdown of cellulose and hemicellulose. Is there a phylogenetic relationship of the ant cultivar to brown rot fungi? Or has the trait of producing H_2_O_2_ and the entailing reactions evolved multiple times?

The brown rot and white rot fungi terminology only applies to wood degrading fungi found within the basidiomycete subdivision Agaricomycotina. It is a functional categorization based on the ability to degrade lignin, rather than a phylogenetic categorization. The mechanism used by brown rot fungi to degrade wood is probably the default one, used by lineages that have not developed the white rot mechanism of active lignin degradation or have secondarily lost it. The attine ant fungal symbionts belong to the order Agaricales within the Agaricomycotina and we have shown previously (Nygaard et al., 2016) that the leaf-cutting ant cultivar has lost the (white-rot-analogous) lignin peroxidases that the cultivars of other, phylogenetically more basal, attine ant species have. It thus appears like the leaf-cutting ant cultivar has reverted to a brown rot-like mechanism from a white rot-like mechanism used by its ancestors. It thus seems reasonable to expect that this loss only happened after alternative and possibly more sophisticated mechanisms for lignin degradation had evolved. We have now explained this more explicitly (Results).

2) In brown rot fungi aromatic alcohols play an important role. These alcohols are oxidized in order to produce H_2_O_2_. The aromatic alcohol veratryl alcohol produced by brown rot fungi was also used here in experimental procedures, where the authors showed that H_2_O_2_ is produced in larger quantities when it is present. Is anything known about the concentration of such aromatic alcohols in the fecal fluid of leaf-cutter ants? How would it be produced (and by whom?)?

To our knowledge this issue has never been investigated. As written in the manuscript, we conjecture that the fungal cultivars provide these alcohols, which would explain the upregulation of the phenylalanine/tyrosine pathway in the gongylidia (Fine Licht et al., 2014), the swollen hyphal tips that the ants ingest, because phenylalanine is the precursor of veratryl alcohol. This is clearly a subject that deserves to be looked into in more detail in the future.

3) Is there any evidence for a high concentration of citrate in the gut of leaf-cutter ants? Indirect evidence is presented by the authors (Results) with a Mollicute bacterium that has a citrate catabolizing pathway. However, is this bacterium part of the core microbiome? Any possibility to provide more direct evidence here? The Mollicute bacterium would only be an opportunist scavenging the citrate? In Sapountzis et al., 2018, the information is given that the leaf-cutter workers would consume relatively high levels of citrate with juices from freshly cut plant material. I think that this link may be more convincing.

To our knowledge the concentration of citrate in the guts of leaf-cutting ants has not been measured. We consider Mollicutes as part of the core gut microbiome of leaf-cutting ants, and in addition they are often found as intracellular bacteria in gut cells and Malpighian tubules. It is thus hard to imagine that they would only be opportunist bacteria. Several aspects of the sequenced genomes of these Mollicutes also indicate specialized functions as co-adapted symbionts (cf. Sapountzis et al., 2018). The Mollicute strain found in leaf-cutting ants has a citrate catabolizing pathway that is very rare among Mollicutes, and it thus seems most parsimonious to assume that they have acquired this pathway because they assimilate citrate from the gut lumen. However, the paragraph in question has now been removed because the possible link between Fenton chemistry and citrate catabolism became less important after revising the text on iron content of fecal fluid being surprisingly high.

4) I really like the point of the Discussion where the authors point out that the Fenton balls are spatially separated bioconversion reactors that prevent tissue damage of either ant of fungus when high levels of reactive oxygen is presumably produced.

We also find this an important part of the story.

5) A very interesting point that the authors touch for future studies is that other higher leaf-cutter ants of genus Atta do not seem to make use of mechanism for the breakdown of plant material studied here for Acromyrmex. Is this really a total black box? Or could the authors at least provide an idea how it works in Atta?

We have not looked into this, and would prefer not make unsolicited hypotheses in this manuscript. We suspect that it will take a complete fecal fluid proteome to begin understanding the adaptations that *Atta* must have evolved independently and differently. Although Fenton chemistry made sense after we put all the *Acromyrmex* puzzle pieces together, it would not be surprising if a different innovation of similar or even higher efficiency in *Atta* would turn out to be equally surprising.

6) In contrast, I think the last section of the Discussion is only partially helpful. While I agree with the authors that the high level of physiological complementarity and the large proportion of fungal enzymes in the fecal fluid renders the term “manure” inappropriate I do think that the suggested term “bloodstream” is not very helpful. The superorganism does not have a bloodstream (and also not a joint symbiotic one). I am not sure whether this highly anthropomorphic analogy is helpful in the future. I suggest to tone down this paragraph and stress the functional physiological complementarity more.

We appreciate this comment and can see that the “bloodstream” analogy was not particularly helpful. In fact the reason for adding this paragraph was, precisely as the reviewer writes, to be careful with anthropomorphic language. We have now replaced that term by “higher-order circulation system”, which we hope will be less distracting and help focus on the real issue, which is that ant farming has very few deep analogies with human farming even though established terminology continues to suggests such analogies. The second author now has an “in press” paper working out the details of these semantic issues that deserve to be reappreciated, which has now been added as source. We hope you can agree that ending the paper this way is reasonable (Discussion).

Reviewer #4:[…] 1) In the Introduction, it is stated that saprotrophs are not well adapted to efficiently degrade fresh plant material. I think there are many sparotrophic fungi well-adapted to do this.

What we mean here is that saprotrophic fungi (in contrast to biotrophic and necrotrophic fungi) are not adapted to cope with live plant tissues that are still able to defend themselves against pathogens. If they were able to do that, they would most likely qualify as necrotrophic fungi. It is of course debatable whether the leaf substrate collected by the ants is still able to mount induced anti-herbivory defenses. However, given that many plants can be grown from plant cuttings we believe that cells in the leaf fragments are likely to remain alive for a considerable amount of time after harvest, and would thus probably still be able to produce substances to deter herbivores.

2) Throughout the work, oxidoreductases have been categorized separately from other CAZY enzymes. For example, in the Results and in Table 1 and Figure 1. However, enzymes such as laccase (AA1), GMC oxidoreductase/aryl-alcohol oxidase (AA3), etc. are also classified as CAZY enzymes, although as having auxiliary activities. Could oxidoreductases be structured as a sub-category of CAZY enzymes to distinguish them from hydrolytic enzymes?

It is correct that oxidoreductases are listed in the CAZy database in the category Auxiliary Activities. However, in the description of this category it is stated that it covers “redox enzymes that act in conjunction with CAZymes”, which implies that they are not CAZymes themselves. Also, since some of the oxidoreductases listed in Table 1 are not covered by the CAZy database, we find it most appropriate to keep them separate from the classical carbohydrate active enzymes.

We have now added the following sentence in the legend of Table 1 to clarify this point: “Although a number of oxidoreductases are listed in the CAZy database in the subcategory of Auxiliary Activities we have for the present study kept them separate. […] This separation also resolved the problem that some of the oxidoreductases listed in Table 1 are not covered by the CAZy database.” In addition we have inserted a note in the Oxidoreductase field in the table indicating that this category includes CAZymes with auxiliary activity.

3) In the Results and Discussion, it is stated that brown rot fungi rely on iron already present in the wood to employ the Fenton reaction. This idea may be somewhat outdated as XRF studies in Serpula lacrymans have shown that this brown rot fungus actively increases the iron concentrations in wood before decomposition. See Kirker et al., 2017. Synchrotron-based X-ray fluorescence microscopy enables multiscale spatial visualization of ions involved in fungal lignocellulose deconstruction. Scientific Reports 7, Article number: 41798.

We were not aware of this interesting publication and thank the reviewer for pointing it out to us. We have now rephrased the relevant sentence to include the findings of this paper (Results).

4) Discussion. I do not see how laccases in the fecal fluid can be considered functional analogs of brown rot recycling mechanisms of Fe(III) to Fe(II) directly. Laccases in the system described by Guillen et al., 2000 oxidise quinones. Reduced quinones in that system are responsible for reducing Fe(III) to Fe(II) and the production of H_2_O_2_. This sentence reads as if the laccases perform this function instead of the quinones.

In the system described by Guillen et al., 2000, laccase is needed to oxidize hydroquinone into semiquinone that is subsequently able to convert Fe3+ to Fe2+. In the manuscript sentence referred to, we did not mention that hydroquinone is used as a substrate in the attine symbiotic reaction as this information had already been given before (Discussion) and was not an essential part of the present argument. However, to avoid misunderstanding we have now changed the sentence to: “The high concentrations of laccase LgLcc1 (Protein ID 97 in Table 1) in the ant fecal fluid suggest a similar mechanism of maintaining reliable conversion of Fe^3+^ back to Fe^2+^ (using hydroquinone as an electron donor) to the one known from brown-rot fungi (Guillen et al., 2000)”.

5) Results. Not only enzymes can reduce Fe(III) to Fe(II) in brown rot fungi. The recycling of Fe(III) to Fe(II) in brown rot systems can be attributed to quinones as well.

As we have mentioned in a response to a previous comment above, it is believed – consistent with Guillen, 2000, – that in natural systems Fe^3+^ will always be bound by metal chelators, which makes Fe^3+^ unavailable to be directly reduced by hydroquinone. It thus needs laccase for this purpose. The affinity of the metal chelators will, however, depend on ambient pH, such that at very low pH it might be possible that Fe^3+^ can be directly reduced by hydroquinone. We prefer, however, not to go into this discussion, as the manuscript is complicated enough already.